# In-depth correlation analysis between tear glucose and blood glucose using a wireless smart contact lens

Wonjung Park [1,2,15], Hunkyu Seo [1,2,15], Jeongho Kim[3,15], Yeon-Mi Hong[1,2], Hayoung Song [1,2], Byung Jun Joo[1,2], Sumin Kim [1,2], Enji Kim [1,2], Che-Gyem Yae[4], Jeonghyun Kim [5], Jonghwa Jin[6], Joohee Kim [7] ✉, Yong-ho Lee [8,9,10] ✉, Jayoung Kim [11] ✉, Hong Kyun Kim [3,4,12] ✉ & Jang-Ung Park [1,2,13,14] ✉

Tears have emerged as a promising alternative to blood for diagnosing diabetes. Despite increasing attempts to measure tear glucose using smart contact lenses, the controversy surrounding the correlation between tear glucose and blood glucose still limits the clinical usage of tears. Herein, we present an in-depth investigation of the correlation between tear glucose and blood glucose using a wireless and soft smart contact lens for continuous monitoring of tear glucose. This smart contact lens is capable of quantitatively monitoring the tear glucose levels in basal tears excluding the effect of reflex tears which might weaken the relationship with blood glucose. Furthermore, this smart contact lens can provide an unprecedented level of continuous tear glucose data acquisition at sub-minute intervals. These advantages allow the precise estimation of lag time, enabling the establishment of the concept called 'personalized lag time'. This demonstration considers individual differences and is successfully applied to both non-diabetic and diabetic humans, as well as in animal models, resulting in a high correlation.

Continuous monitoring of biomedical signals in the human body is imperative for the diagnosis and management of metabolic disorders. With the development of wearable electronics, extensive efforts have been made to monitor diverse physiological signs through physical parameters (e.g., blood pressure, heart rate, electrocardiogram) and metabolites (e.g., glucose, amino acid, vitamins) over the past decade[1–9]. Among various metabolites, glucose is a significant biomarker for the diagnosis and management of diabetes, but it fluctuates

[1]Department of Materials Science and Engineering, Yonsei University, Seoul 03722, Republic of Korea. [2]Center for Nanomedicine, Institute for Basic Science (IBS), Yonsei University, Seoul 03722, Republic of Korea. [3]Cell and Matrix Research Institute, School of Medicine, Kyungpook National University, Daegu 41944, Republic of Korea. [4]Department of Ophthalmology, Kyungpook National University School of Medicine, Daegu 41944, Republic of Korea. [5]Department of Electronics Convergence Engineering, Kwangwoon University, Seoul 01897, Republic of Korea. [6]Department of Internal Medicine, School of Medicine, Kyungpook National University, Kyungpook National University Hospital, Daegu 41944, Republic of Korea. [7]Center for Bionics, Biomedical Research Division Korea Institute of Science and Technology, Seoul 02792, Republic of Korea. [8]Department of Internal Medicine, Yonsei University College of Medicine, Seoul 03722, Republic of Korea. [9]Institute of Endocrine Research, Yonsei University College of Medicine, Seoul 03722, Republic of Korea. [10]Institute for Innovation in Digital Healthcare (IIDH), Severance Hospital, Seoul 03722, Republic of Korea. [11]Department of Medical Engineering, Yonsei University College of Medicine, Seoul 03722, Republic of Korea. [12]Department of Ophthalmology, Kyungpook National University Hospital, Daegu 41944, Republic of Korea. [13]Department of Neurosurgery, Yonsei University College of Medicine, Seoul 03722, Republic of Korea. [14]Graduate Program of Nano Biomedical Engineering (NanoBME), Advanced Science Institute, Yonsei University, Seoul 03722, Republic of Korea. [15]These authors contributed equally: Wonjung Park, Hunkyu Seo, Jeongho Kim. ✉e-mail: joohee710610@kist.re.kr; yholee@yuhs.ac; jayoungkim@yonsei.ac.kr; okeye@knu.ac.kr; jang-ung@yonsei.ac.kr

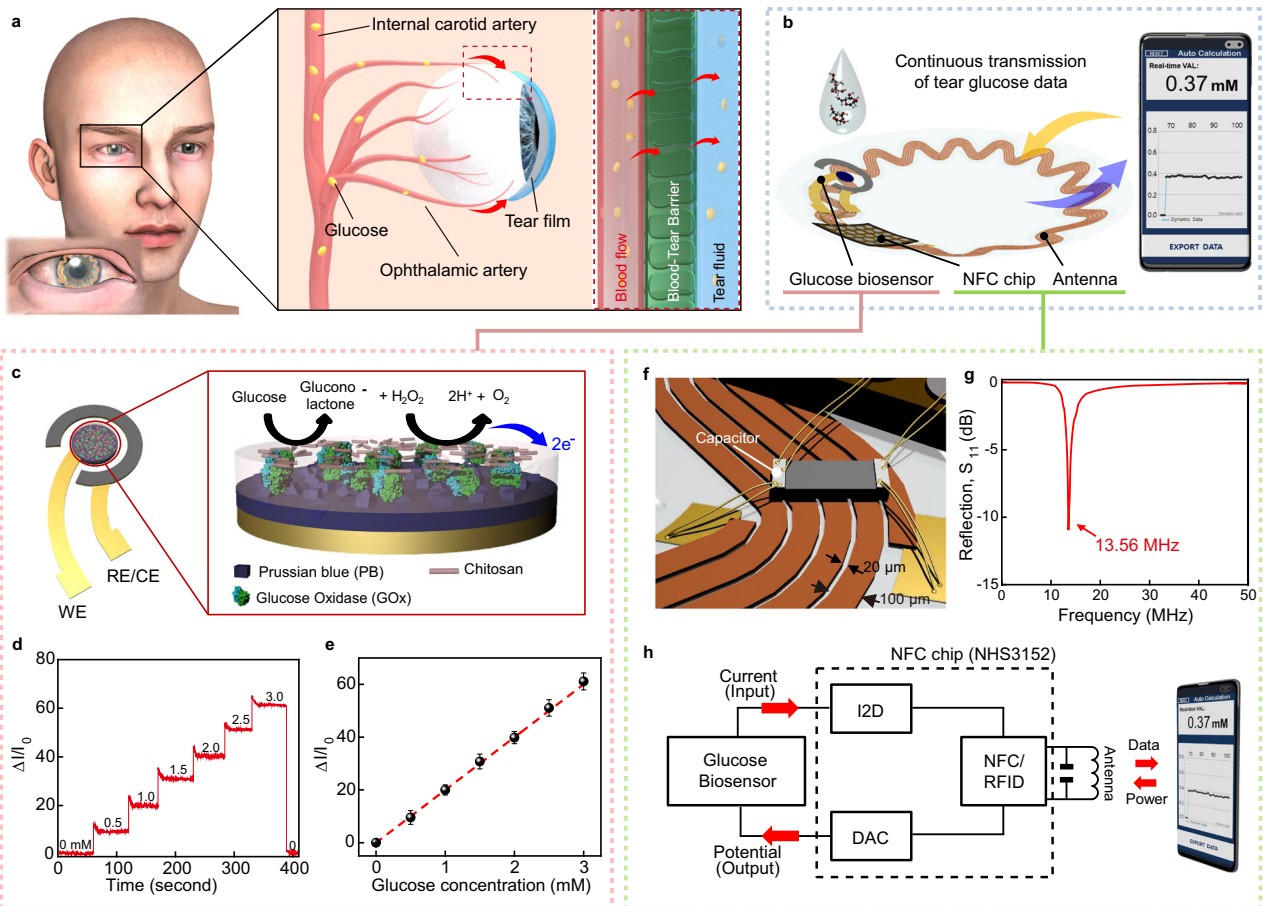

**Fig. 1 | Characterization of a smart contact lens for monitoring tear glucose. a** Schematic illustration of a plasma leakage in human eyes. Red arrows indicate the leakage of glucose from blood into the tear film. **b** Schematic illustration of the tear glucose (TG) monitoring system using the smart contact lens (SCL) and smart phone. **c** Schematic illustration of the electrochemical glucose biosensor with two electrodes and magnified electrochemical reaction layer of working electrode. **d** Relative changes in current at various concentrations of glucose. **e** Calibration curve of the relative changes in current as a function of glucose concentration ($n = 20$). Each data point represents the average of 20 samples, and the error bars indicate mean ± standard error of the mean. **f** Schematic illustration of the serpentine antenna and the capacitor integrated to the antenna pad. **g** The resonance frequency of the serpentine antenna integrated into the SCL. **h** Circuit diagram of the SCL system for the wireless monitoring of the glucose level.

drastically depending on the patient's diet and lifestyle. Thus, continuous glycemic control is important to prevent the microvascular complications[10], e.g., neurological, retinal, and renal complications, that can occur with diabetes. However, the measurement of glucose by a glucometer involves invasive blood sampling, and it has the limitation of single-time-point measurements, making compact glycemic management impossible. To overcome this limitation, the continuous measurement of glucose has been conducted with various body fluids, e.g., tears[11,12], sweat[13,14], saliva[15], and interstitial fluid (ISF)[16,17].

Tears, one of the most promising body fluids as a non-invasive health indicator, also contain glucose, and higher levels of tear glucose (TG) have been observed in diabetic patients compared to normal individuals[18,19]. The most compelling reason for the presence of glucose in tears can be explained by a phenomenon called "plasma leakage" which refers to the partial leakage of the components of blood into tear fluids through the blood-tear barrier[20,21] (Fig. 1a). Since the feasibility of diagnosing diabetes through tears has been demonstrated, there has been extensive interest in the development of wearable devices that can monitor TG levels[22–25]. Revealing the relationship between TG and BG is crucial to evidence the usefulness of TG as a noninvasive alternative of BG, and significant efforts have been made, as summarized in Supplementary Table 1; however, the correlation between these two still remains controversial[12,18–20,22–24,26–39]. A strong candidate for the cause of this controversy is the variation in the

composition of tears depending on the tear collection method. The commonly used methods (e.g., filter paper strip, Schirmer strip, capillary tube) can stimulate the eye to generate stimulated tears (i.e., reflex tears) having different tear compositions, including tear glucose levels, compared to non-stimulated and retained tears (i.e., basal tears)[40,41], restricting reliable and reproducible tear collection and TG measurement. In addition, these conventional methods usually allow single measurement of TG, limiting continuous tracking of TG changes, and further failing to demonstrate their relationship with dynamic BG behaviors along with lag time. Lag time is the time for BG to diffuse into tears, and it is an important factor determining the degree of correlation between TG and BG[32]. The accurate estimation of the lag time can be obtained by the continuous and frequent measurement of the dynamic changes of TG and BG. A contact lens platform that is capable of non-invasive and continuous monitoring of ocular biomarkers can overcome these limitations that are associated with the collection of tears[11,42–47]. Contact lenses that are worn on the eyes do not require an operator's skills for the collection of tears, and they can be in direct contact with basal tears, and this allows continuous analysis of the tear fluid. Furthermore, due to their soft, stretchable, and biocompatible properties, contact lens can provide comfortable wearing and minimize eye irritation. Therefore, the contact lens platform can be an advantageous method for analyzing the correlation between TG and BG. Early efforts have been made on TG monitoring

using smart contact lenses (SCLs), showing the feasibility of TG detection at a proof-of-concept level, and most of them were limited to single-time-point measurement[22,23,33]. The recent advances, showing high performance on TG monitoring using SCL by incorporating bimetallic nanocatalysts, demonstrated periodic and repetitive measurements of TG for the correlation analysis between TG and BG, applying lag time; however, they still had limitations in analytical methods. In detail, the sparse 5-min data acquisition interval of TG may restrict the accurate identification of lag time, since glucose level change can be very instantaneous, and 10 min of the estimated lag time were applied universally to all animals and subjects without considering the differences in glucose metabolisms among individuals. Moreover, each type of glucose level dynamics, such as increasing, decreasing, and maintaining stages, was demonstrated as different sets of experimental trials, and this could not mimic the practical scenario that usually occurs in a series of episodes. Therefore, the SCL studies for measuring glucose that have been performed to date are still limited in terms of providing a proper correlation analysis of TG with BG for accurate prediction of BG using TG.

Herein, we developed a wireless and soft SCL glucose biosensor and employ it to demonstrate monitoring TG as a noninvasive alternative of BG by addressing existing concerns and suggesting new methodologies for precise correlation analysis between BG and TG. First, we investigated whether SCL induces reflex tearing in rabbit models since the generation of reflex tears has been one of the major reasons for raising doubts about the reliability of TG for BG prediction. In addition, to further investigate the impact of reflex tears on TG levels, external mechanical stimulations were applied to the conjunctiva while wearing SCL, and the changes in the TG levels were evaluated. Second, we established the concept of "personalized lag time" and applied to all of the subjects for the precise correlation analysis between TG and BG. Because individuals have different rates of carbohydrate metabolism, it is important to consider the individual lag times. The developed SCL glucose biosensor allowed frequent and long durations of continuous measurements of TG, enabling the precise identification of the lag time in each individual. Each personalized lag time was applied to all individuals for in-depth correlation analysis of TG and BG. Lastly, such correlation analysis using the developed SCL was further demonstrated with healthy and diabetic human participants, and different animal species (e.g., herbivorous rabbits and omnivorous beagles) with and without diabetes for comparative analysis. For the better prediction of BG, the above-mentioned concept of personalized lag time has been utilized successfully in humans, rabbits, and beagles which resulted in a clear demonstration of the correlation between TG and BG. Conducting a comprehensive analysis across various species adds further significance for the demonstration of the correlation between TG and BG. To the best of our knowledge, this is the first in-depth study to use SCL for examining the correlation between TG and BG across various species including humans, rabbits, and beagles with diabetes, in comparison with healthy groups as control ones.

Overall, we utilized the developed SCL glucose biosensor to address existing doubts about the reliability of using TG for the prediction of BG, and introduced "personalized lag time", which can be a key factor for the accurate estimation of BG. Such effort can contribute to convincing other concerning the clinical utility of SCL glucose biosensors as biomedical devices for the diagnosis of diabetes and the monitoring of the related treatment.

## Results

### Wireless and soft smart contact lens system for tear glucose monitoring

Our SCL is designed to provide the ability to monitor the glucose level in tears and to transmit this information wirelessly to the user's smartphone. Figure 1b shows the overall schematic illustration of this real-time glucose monitoring SCL system, composed of a glucose sensor and wireless communication components, in the form of a soft contact lens. Among the standard wireless communication technologies, near-field communication (NFC) can be suitable for the smart contact lens because it allows a wireless supply of power to a sensor for its battery-free operation, and it allows the wireless transmission of data with sufficient bandwidth. The detailed fabrication procedure is described in Supplementary Fig. 1 and the Methods section. After locating the stretchable integration of all of the components of the device on a flat elastic film, the molding of the resulting flat sample (with devices) into the curved lens shape by injecting a precursor of silicone elastomer (Interojo®), which is a commercially available material for a soft contact lens, completed the fabrication of our SCL. During this molding step, the selective area of the glucose sensor was open and locally uncovered by the lens material to allow its physical contact with tears. In this way, all of the electronic devices could be embedded inside the soft contact lens with this opening remaining for the sensor (Supplementary Fig. 2). The NFC system of our SCL was able to wirelessly transmit the measured glucose data to a smartphone. The smartphone could display the measured TG concentration in real-time through the smartphone application, and the data were stored in a database.

The SCL was designed for the continuous measurement of the TG concentration using the electrochemical system of the glucose sensor. Glucose oxidase (GOx) was immobilized along with chitosan on the working electrode (WE), and it reacted with glucose to generate gluconolactone and $H_2O_2$ as products (Fig. 1c). Prussian blue (PB) on WE acted as an artificial peroxidase that facilitates the reduction of $H_2O_2$ that resulted from the reaction between glucose and GOx[48]. For the amperometric detection of glucose, a potential of $-0.1$ V (vs Ag/AgCl) was chosen based on the cyclic voltammetry (CV) results in Supplementary Fig. 3. As shown in Fig. 1d, e, the relative change in the current response was measured at various glucose concentrations. The concentration of glucose increased linearly with a sensitivity of 1% change in reduction current per 0.047 mM and a detection limit of 0.02 mM, demonstrating that this developed glucose biosensor could detect glucose variations in human tears (0.18–0.7 mM)[34]. This device exhibited good selectivity in the presence of physiologically relevant interferences (Supplementary Fig. 4a), and it showed negligible differences in various physiological tear pH values[49], from 6.4 to 7.6 (Supplementary Fig. 4b). Also, it provided 21 days of long-term stability at room temperature (22 °C) (Supplementary Fig. 4c).

Remote monitoring of the real-time glucose concentration was performed through the NFC function between SCL and a smartphone, with the stretchable integration of a commercial NFC chip (NHS 3152, NXP Semiconductors, thickness: 200 μm after polishing), the serpentine antenna, capacitor, and the glucose sensor. To match the standardized resonance frequency of NFC (13.56 MHz), as illustrated in Fig. 1f, an 81 pF capacitor was integrated into the stretchable serpentine antenna that was designed for this resonance frequency based on calculations using the High-Frequency Structure Simulator (HFSS) program. The resulting SCL exhibited a resonance frequency centered at 13.56 MHz (Fig. 1g). When the antenna supplies power to the wireless circuit through inductive coupling with a smartphone, a Digital-to-Analog Converter (DAC) in the NFC chip applied a constant voltage ($-0.1$ V) to the glucose sensor for the electrochemical sensing (Fig. 1h). The resulting current response was read by the Current-to-Digital converter (I2D) with 100 ms integration time, which was sufficient to monitor the changing TG, and the current data were transmitted wirelessly to the smartphone. In a smartphone application, the concentration of glucose calculated from the current value that is detected could be updated every second.

Successful demonstration of the integrated SCL biosensor on the eye of a mannequin was performed for wireless glucose sensing with a smartphone, and it showed only 3% error or less with the reference

concentration of glucose (Supplementary Fig. 5 and 6). The biocompatibility of SCL was evaluated in human corneal cells (HCE-2) and human conjunctival cells (HCECs) using a LIVE/DEAD assay kit and Cell Counting Kit-8 (CCK-8). Supplementary Fig. 7 shows that 97.88 ± 1.82% of HCE-2 and 95.28 ± 1.67% of HCECs did not exhibit significant cytotoxicity compared to controls. Calcein-AM/ethidium homodimer-1 double staining also revealed no significant dead cells, confirming negligible cytotoxicity (Supplementary Fig. 8). These results indicate that our SCL satisfied the cytotoxicity standard of over 80% for medical devices (ISO 10993-5).

### Effects of reflex tears on the correlation between TG and BG

There has been a continuing controversy about the clinical significance of TG as a biomarker for diabetic mellitus, especially, due to variations in tear composition caused by physical stimulation of the eye, which can induce reflex tears and cause fluctuations in TG. For example, conventional methods of collecting tears (e.g., filter paper and Schirmer strip) elicit reflex tears because these sharp tips come into direct contact with the ocular surface. However, the SCL is a relatively soft, stretchable, and biocompatible device that can continuously and comfortably contact tear fluid, and it is expected to minimize tear stimulation when worn. As shown in Fig. 2a, the changes in tear volume (TV) and TG were monitored continuously after our SCLs were worn on the eyes of four normal rabbits. Here, TG was measured through the SCL every 20 s from the time of wearing the SCL, and TV was measured every minute through a strip meniscometry tube (SMTube) by absorbing tear film meniscus (Supplementary Fig. 9a). As plotted in Fig. 2b, c, when the SCL was worn, both TV and TG levels immediately increased (due to reflex tearing) and then reduced rapidly again close to their initial states. These rapid recoveries of both TG and TV appeared about 160 and 135 s, respectively, after wearing the SCL (Fig. 2d). Then, TG and TV became stabilized without significant fluctuations, indicating negligible additional eye irritation from wearing the SCL. The disturbances of the TG and TV levels only occurred for a short time when the SCL was being worn. Therefore, in all of the following in vivo experiments, TG was measured by giving sufficient stabilization time of 10 min after wearing the SCL.

Further, we investigated the effect of physical stimulation-induced reflex tears on the correlation between TG and BG in rabbit models while wearing the SCL. As shown in Fig. 2e, continuous measurements of the levels of BG, TG, and TV were performed in four normal rabbits with mechanical eye stimulation. For the external stimulation to the conjunctiva, the von Frey filament, which can provide constant calibrated force, was used to induce the additional reflex tear fluid (Fig. 2f and Supplementary Fig. 9b). The strength of the stimulation force that can induce sufficient reflex tearing while minimizing conjunctival damage was determined to be 1.4 gf, and no significant increase in TV occurred with stronger forces (Fig. 2g and Supplementary Fig. 10). Figure 2h is the resulting representative plot among four normal rabbits. After stimulation on the SCL-worn conjunctiva, TV increased rapidly, along with a TG spike. It is consistent with the findings from previous studies that mechanical stimulation can cause glucose to leak directly from the interstitial space or epithelial cells into the tear fluid[20,41]. After the spike, both TG and TV soon recovered. The same trend of variations was observed in the other three rabbits (Supplementary Fig. 11). As shown in Supplementary Fig. 12, the average recovery times for TG and TV in the four rabbits were ~300 s and 260 s, respectively. The recovery time refers to the time taken from the secretion of reflex tears to the complete clearance of the effects of the reflex tears due to the repeated secretion and drainage processes of the tears (i.e., tear circulation)[50]. It implies that, although TG measurement after this stimulation can compromise its correlation with BG, it can be recovered soon. Note that SCL allows us to obtain stable TG values in a continuous way while wearing and when external stimulation is not present, and even when it occurs unexpectedly, TG can

recover soon and allows us to obtain reliable TG values. Such a feature enables the in-depth investigation of the correlation between TG and BG in a continuous manner with in vivo models.

### In vivo correlation analysis between BG and TG with normal and diabetic rabbit models

In this section, we present the availability of SCL for TG monitoring and the establishment of a correlation analysis method between BG and TG by measuring both in a real-time and continuous way via the oral glucose tolerance test (OGTT) in normal rabbits, as well as via the intravenous glucose tolerance test (IVGTT) in normal and diabetic rabbits (Fig. 3a and Supplementary Movie 1). Initially, we conducted studies to validate the reliability of SCL and to establish a correlation analysis method using a normal rabbit model. In these studies, rabbits with fasting were anesthetized to minimize the changes in TG through additional movement or stimulation. When the BG level was stable, the TG level of both eyes showed a normal range of 0.45–0.48 mM without any significant variations, indicating that the TG level of one of both eyes can represent the actual TG level of the rabbit if there is no additional stimulation of the eyes (Supplementary Fig. 13). Next, an OGTT was conducted on normal, fasting rabbits to evaluate the relationship between TG and BG by observing variations in glucose levels. A 50% (w/w) glucose-concentrated solution at 3.0 g kg$^{-1}$ of body weight was orally administered to four normal rabbits. The BG level was measured at 10-min intervals using a glucometer concurrently with continuously measuring TG using the SCL (Fig. 3b). In the fasting state, rabbit subject 1 showed 0.423 mM of TG and 6.21 mM of BG, which were within the normal range of TG[36] and BG[51] for rabbits. After oral administration of the glucose solution, the TG and BG levels increased gradually and showed the highest levels of 1.059 mM and 11.04 mM respectively, and then the levels decreased gradually to 0.614 mM and 6.58 mM, respectively, under the influence of insulin via glucose homeostasis[52] (Fig. 3c). All four rabbits showed similar trends of up-and-down in both TG and BG levels, but each rabbit had different values of TG and BG. These different values were due to the variations in the glucose metabolism and metabolic actions of insulin among individual rabbits even though they had consumed the same amount of glucose per weight (Supplementary Fig. 14). Also, TG values reflected the trend of BG with certain delayed time in all four rabbits. Such phenomena often occur in non-invasive biofluids, such as sweat, and saliva and the delayed time corresponds to the time for the biomarkers to diffuse from the blood vessels to the biofluids[53–55]. We called it "lag time" and calculated it for each rabbit to identify how it can impact Pearson's correlation coefficient between BG and TG. The correlation coefficient and regression line between TG and BG levels varied depending on the value of the applied lag time (Supplementary Fig. 15). As shown in Fig. 3d, the correlation coefficient was plotted further with the lag time ranges from 0 to 25 min in increments of 1 min, demonstrating that accurate estimation of lag time is crucial for the higher degree of correlation between TG and BG. The coefficient at 14 min showed the highest value of 0.987, indicating that 14 min was the most precise lag time for the tested rabbit. In this way, the determination of personalized lag times for the other three rabbits was performed, showing 14, 13, and 13 min, respectively (Supplementary Fig. 16). All measured OGTT data were plotted together for four rabbits by applying the personalized lag time for each rabbit. This resulted in 0.944 of Pearson's correlation coefficient, indicating a strong linear relationship between TG and BG (Supplementary Fig. 17).

We further investigated the correlation between TG and BG via IVGTT in normal and diabetic rabbits. The diabetic rabbits were prepared by streptozotocin (STZ) treatment, which is fatal to the insulin-producing beta cells of the pancreas. Over the course of inducing diabetes, the rabbits lost weight and had increases in their fasting BG values (Supplementary Fig. 18). To confirm the induction of diabetes in the rabbit models, we performed histological analysis of the pancreatic

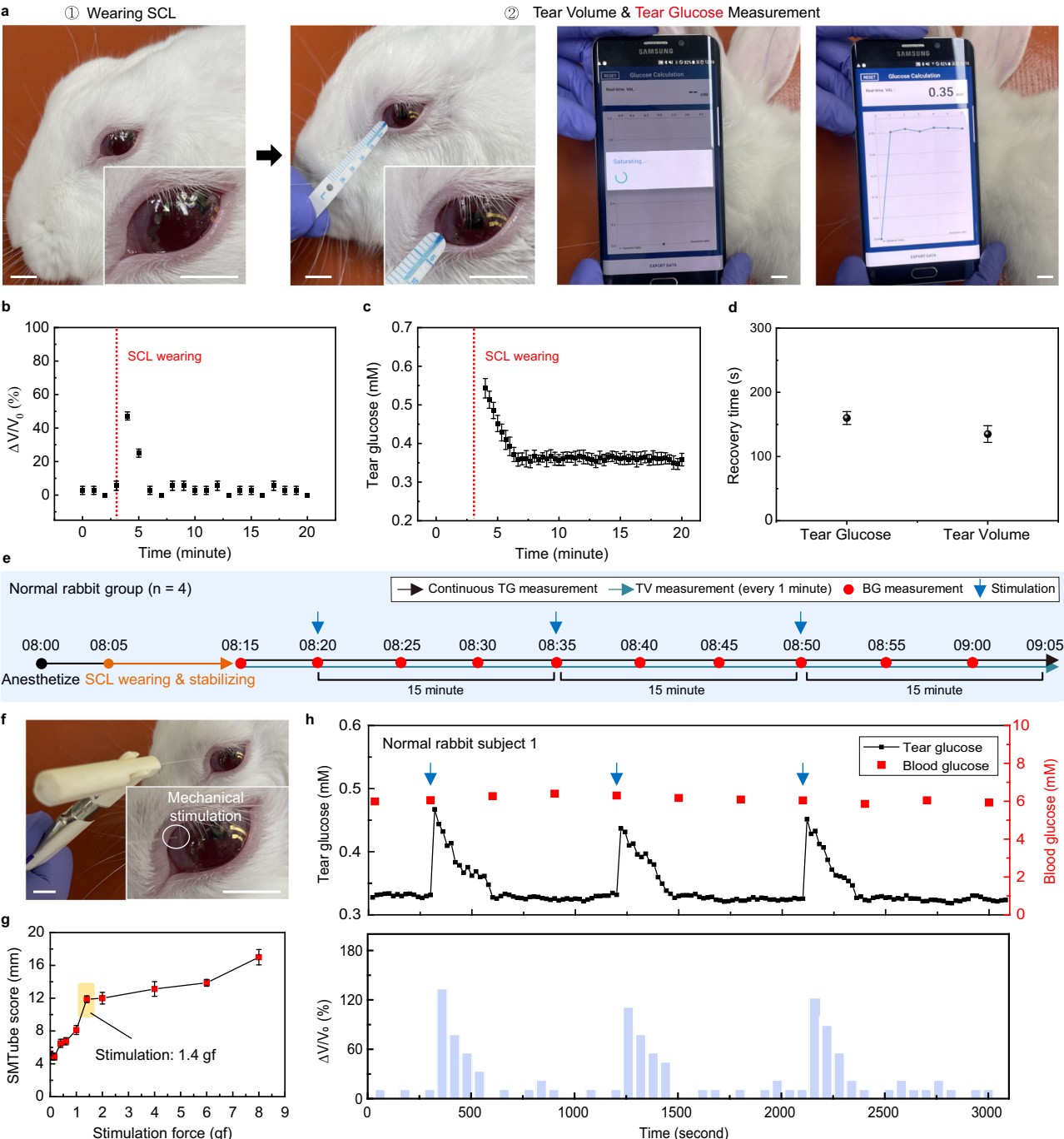

**Fig. 2 | Effect of reflex tears on tear glucose (TG) and tear volume (TV) level.** **a** Photograph showing the sequence for TG and TV monitoring after wearing smart contact lens (SCL). Scale bars, 1 cm. **b** Real-time measurement of TV when wearing SCL in normal rabbits ($n = 4$). **c** Real-time measurement of TG when wearing SCL in normal rabbits ($n = 4$). **d**, Comparison of TG level recovery time and TV recovery time after SCL wearing ($n = 4$). For (**b**–**d**) each data point represents the average of four normal rabbits, and the error bars indicate mean ± standard error of the mean. **e** Timeline of continuous measurement of blood glucose (BG), TG and TV along with conjunctival stimulation. **f** Photograph of external stimulation to the conjunctiva of rabbit using a von Frey filament. Scale bars, 1 cm. **g** Average of SMTube scores of four rabbits as a function of stimulating force ($n = 4$). Each data point represents the average of four normal rabbits, and the error bars indicate mean ± standard error of the mean. **h** Representative data of monitoring TG and BG levels (top) and relative changes in TV along with mechanical stimulation (bottom).

islets after completing the in vivo experiments. Supplementary Fig. 19 displays immunostaining images for glucagon and insulin in the pancreatic islet. The pancreatic islet area of diabetic rabbits was reduced by -97.70% compared to normal rabbits, indicating successful induction of diabetes in rabbit models. In detail, the normal rabbits had an area of $766.41 \pm 150.55\,\mu m^2$, whereas the diabetic rabbits had an area of $17.63 \pm 8.70\,\mu m^2$.

After anesthetizing the 12-h-fasted rabbits, a 50% concentrated glucose solution, at a dosage of $1.0\,kg^{-1}$ of body weight, was injected intravenously through the marginal ear vein. The IV injection of glucose can minimize the variables of glucose absorption rate that may occur in oral administration and can reduce urinary loss of glucose and maximize insulin response[56]. Therefore, the IV injection method enables the clear and effective spike of the glucose in the animal

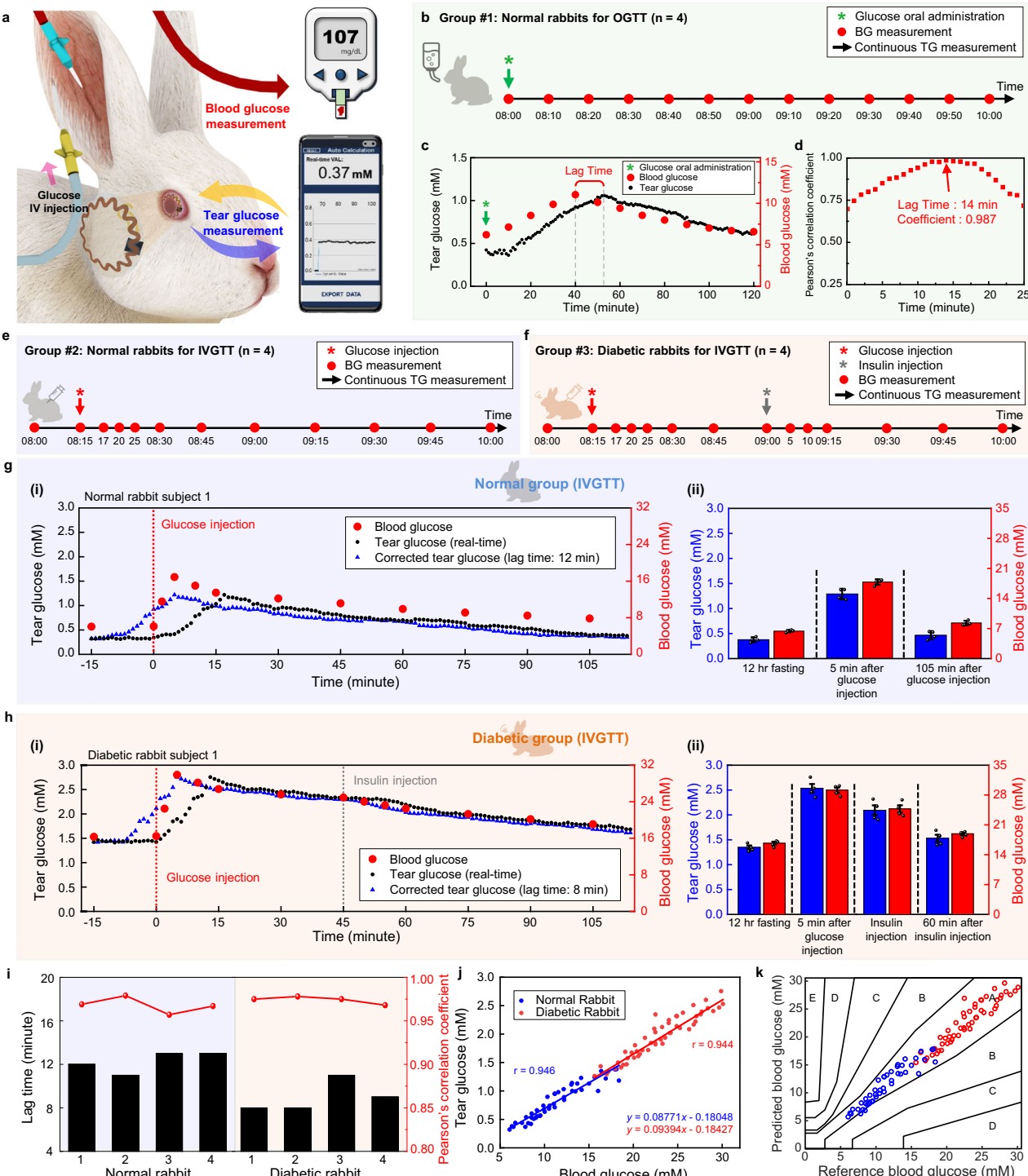

**Fig. 3 | In vivo correlation analysis between tear glucose (TG) and blood glucose (BG) with normal and diabetic rabbits. a** Schematic illustration of the simultaneous measurement of TG and BG during IVGTT in a rabbit model. **b** Timeline of BG, and TG measurement after oral administration of glucose in normal rabbits. **c**, Representative data of monitoring TG and BG in normal rabbits after oral administration of glucose. **d** Pearson's correlation coefficient between TG and BG as a function of lag time. **e** Timeline of continuous measurement of TG and BG after glucose intravenous (IV) injection for normal rabbits. **f** Timeline of continuous measurement of TG and BG after IV injection of glucose and insulin for diabetic rabbits. **g** Representative real-time data of TG and BG in a normal rabbit after IV injection of glucose. (i), Real-time monitoring of TG (black dots), BG (red dots), and corrected TG data (blue dots) of a normal rabbit. (ii), Comparison between the

average of TG and BG in four normal rabbits before and after the injection of glucose, and the error bars indicate mean ± standard error of the mean. **h** Representative real-time data of TG and BG in a diabetic rabbit after IV injection of glucose and insulin. **i** Real-time monitoring of TG (black dots), BG (red dots), and corrected TG data (blue dots) of a diabetic rabbit. (ii), Comparison between the average of TG and BG in four diabetic rabbits before and after injection of glucose and insulin, and the error bars indicate mean ± standard error of the mean. **i** Lag time identification through the Pearson's correlation coefficient in rabbit models. **j** Comprehensive Pearson's correlation analysis of 4 normal rabbits and FOUR diabetic rabbits. **k** Comprehensive Parkes error grid analysis of four normal rabbits and FOUR diabetic rabbits.

models and offer the advantage of systematic analysis of correlation in TG and BG. After wearing the SCL, TG was monitored continuously, and BG was measured through the auricular artery at the designated time (Fig. 3e). An IV injection of glucose was performed via the marginal ear vein. In diabetic rabbits only, 0.5 U kg⁻¹ of insulin was injected 45 min after glucose injection to lower their glucose levels (Fig. 3f). In the case of the normal rabbit model, a representative result among four normal rabbits is shown in Fig. 3g-(i). Normal rabbit subject 1 showed a BG level of 6.06 mM in its fasting state, which was within the BG range of normal rabbits, and it increased to 16.89 mM within 5 min after the glucose IV injection. Subsequently, it decreased gradually to 7.84 mM at 105 min after glucose injection due to glucose homeostasis. Simultaneously, TG also showed a normal fasting level at 0.321 mM and increased to 1.22 mM with glucose injection. Further, it also decreased to 0.369 mM at the end of the experiment, following the dynamic behavior of BG with a certain lag time. Similar trend also was observed in the other three normal rabbits (Supplementary Fig. 20). The lag time of each rabbit between TG and BG was determined as 12, 11, 13, and 13 min, respectively. After the correction of TG data with the lag time value (blue dot), the tendency of TG appeared very similar to the change in the BG level. Figure 3g-(ii) shows the summarized average values for BG and corrected TG from all four rabbits through the course of IGVTT for normal rabbits. For the diabetic rabbit model, a representative result among four diabetic rabbits is shown in Fig. 3h-(i). Despite being in a fasting state, it showed an abnormally high BG level of 16.32 mM, confirming the difficulty in maintaining glucose homeostasis due to damage in beta cells, and further increased to 29.89 mM after 5 min of glucose IV injection. The BG level was then progressively reduced by secreted insulin, which was not as effective as in normal rabbits, but remained much higher than the normal range. Insulin was injected 45 min after the glucose injection to further lower the glucose level in the blood. The BG level started to decrease at a faster rate to 19.09 mM after 60 min of insulin injection, but still out of the normal BG range. The TG level started at 1.434 mM in the fasting state, spiked up to 2.755 mM, and then recovered to 1.687 mM at the end. TG also showed a similar trend to BG with a certain lag time, and the other three diabetic rabbits also exhibited the same trend (Supplementary Fig. 21). The corrected TG data (blue dot) by applying 8, 8, 11, and 9 min of personalized lag time to each diabetic rabbit showed high correspondence with BG. Figure 3h-(ii) presents the average values of corrected TG and BG in these four diabetic rabbits. Figure 3i displays the difference in the lag time among all of the rabbit models. From these results, we concluded that personalizing the lag time is crucial and plays an essential role in accurate identification of the correlation between TG and BG. Pearson's correlation coefficient, calculated by plotting all measured values from 8 rabbits, was high enough with the values of 0.946 in the normal group and 0.944 in the diabetic group (Fig. 3j). We introduced Parkes Error Grid Analysis to evaluate the accuracy of the predicted BG based on TG. The Parkes Error Grid is a commonly used method to assess the clinical accuracy of predicted BG, and reasonable predictions can be made when the predicted BG belongs to Zones A and B[57]. As presented in Fig. 3k, all plotted data points from normal (blue dots) and diabetic (red dots) rabbits were located in the A and B regions, which represent that the prediction of BG level with measured TG data can be clinically accurate.

**In vivo correlation analysis between BG and TG with normal and diabetic beagle models**
To further analyze the correlation between the TG and BG levels across different species, an IVGTT with SCL was conducted on the beagle model (Fig. 4a, b, and Supplementary Movie 2). The beagle is an omnivorous animal that has a diet pattern that is more similar to humans than rabbits. Therefore, conducting correlation analysis using the beagle model may provide more precise predictions for the human

model. Beagles were injected with STZ to induce diabetes. Induction was confirmed by changes in body weights (Supplementary Fig. 22) and histological analysis after all experiments. Supplementary Fig. 23 shows a reduction in pancreatic islet area of 86.51% in diabetic beagles compared to normal beagles (normal beagles: 1506.51 ± 271.80 μm², diabetic beagles: 203.17 ± 70.11 μm²), indicating successful induction of diabetes.

After wearing SCL, TG was monitored continuously, and BG was measured via the cephalic vein at the designated time points (Fig. 4c). The IV injection of glucose was conducted via the foreleg vein after anesthetizing. For diabetic beagles only, 0.5 U kg⁻¹ of insulin was injected 45 min after glucose injection (Fig. 4d). As a result of IVGTT, Fig. 4e-(i) shows the representative TG and BG data for a normal beagle subject 1. The BG levels increased rapidly from a baseline of 3.94 mM, which was within the reported normal range for beagles[32], then reached a maximum value of 16.72 mM at 5 min after the glucose injection. Subsequently, it gradually decreased to 5.67 mM approaching the normal range. In the case of TG, it was at a fasting level of 0.24 mM showing a similar value with the reported normal beagle's TG range[32]. TG further increased to 0.91 mM after the IV glucose injection and then decreased to 0.279 mM after 105 min due to the action of insulin. Similar results were observed in the other three normal beagles (Supplementary Fig. 24). The lag time derived from the measured data of each normal beagle was 12, 10, 9, and 10 min, respectively. The corrected TG data (blue dot) appeared very similar to the BG data. Figure 4e-(ii) presents the summary of IVGTT by showing the average values of the corrected TG and BG from four normal beagles. A representative result among four diabetic beagles is shown in Fig. 4f-(i). An abnormally high value of fasting BG level was shown as 13.83 mM, proving successful induction of diabetes. The maximum level of 28.39 mM was observed 5 min after glucose IV injection. Then, the BG level started to decrease continuously due to glucose homeostasis, but it was not effective enough to bring the BG level to the normal range. Therefore, additional insulin was administered 45 min after the glucose injection to lower the BG level. Subsequently, the glucose continued to decrease and reached the normal range of 5.11 mM after approximately 105 min[33]. In the case of TG, the fasting value was 0.75 mM, which then increased to maximum level of 2.03 mM, and further decreased to 0.58 mM, following the trend of BG with a certain lag time. And this tendency also was observed in each of the other three diabetic beagles (Supplementary Fig. 25). The corrected TG data (blue dot) applied 9, 8, 10, and 7 min of lag time to each diabetic beagle also appeared very similar to the BG data. As shown in Fig. 4f-(ii), the average values of the corrected TG and BG in four diabetic beagles were summarized. Figure 4g shows that lag time differences also appeared between individual beagles. The resulting Pearson's correlation coefficients are 0.924 and 0.961 in the normal and diabetic groups, respectively (Fig. 4h). In the Parkes Error Grid, all data points for normal (blue dots) and diabetic (red dots) beagles are located in the A and B regions, indicating that the prediction of BG level with measured TG data is clinically correct and shows high predictive accuracy (Fig. 4i).

Animal species are classified into carnivores, herbivores, and omnivores based on their diet, and it is known that glucose metabolism differs depending on the species[58,59]. To investigate the correlation between TG and BG among animal species, we conducted a comparative analysis with the experimental results of rabbits and beagles. There were differences in the lag time among individuals in all models, highlighting the importance of identifying the personalized lag time of each animal for proper correlation analysis. In addition, the lag time values in the diabetic model for both rabbits and beagles were relatively shorter than those in the normal model, regardless of individual differences or species. Diabetes can increase tear osmolarity, alter ion transport, and affect blood flow rate around the eyes, thereby promoting glucose diffusion across the blood-tear barrier[20,60,61].

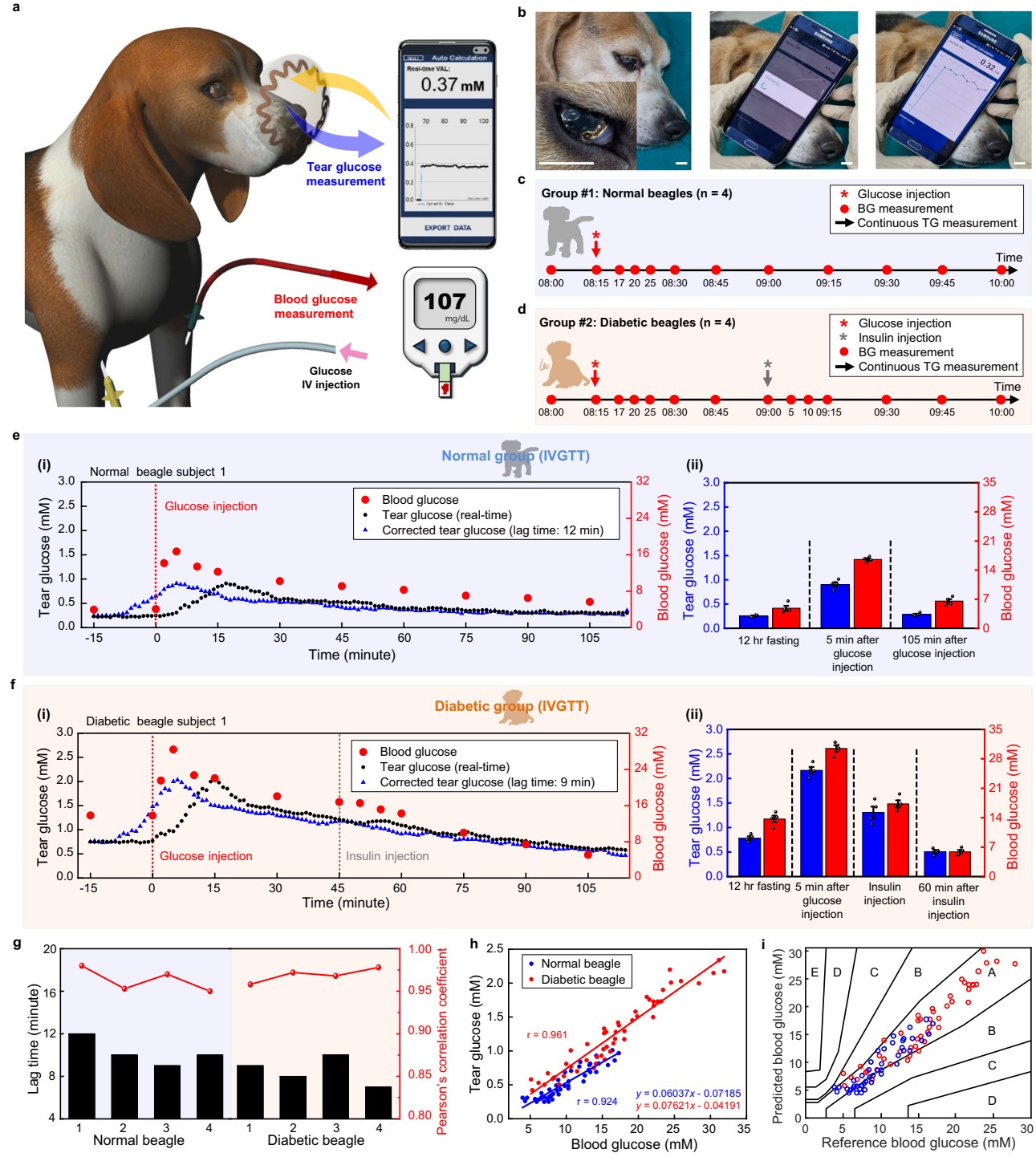

**Fig. 4 | In vivo correlation analysis between tear glucose (TG) and blood glucose (BG) with normal and diabetic beagles. a** Schematic illustration of simultaneous measurement of TG and BG during intravenous glucose tolerance test (IVGTT) in a beagle model. **b** Photographs of simultaneous measurement of TG measurement via SCL. Scale bars, 1 cm. **c** Timeline of continuous measurement of TG and BG at designated time after glucose intravenous (IV) injection in normal beagles. **d** Timeline of continuous measurement of TG and BG at designated time after IV injection of glucose and insulin in diabetic beagles. **e** Representative real-time data of TG and BG in a normal beagle after IV injection of glucose. (i), Real-time monitoring of TG (black dots), BG (red dots), and corrected TG data (blue dots) of a normal beagle. (ii), Comparison between the average of TG and BG in four normal beagles before and

after injection of glucose, and the error bars indicate mean ± standard error of the mean. **f** Representative real-time data of TG and BG in a diabetic beagle after IV injection of glucose and insulin. (i), Real-time monitoring of TG (black dots), BG (red dots), and corrected TG data (blue dots) of a diabetic beagle. (ii), Comparison between the average of TG and BG in four diabetic beagles before and after injection of glucose and insulin, and the error bars indicate mean ± standard error of the mean. **g** Lag time identification through the Pearson's correlation coefficient of every beagle models.
**h** Comprehensive Pearson's correlation analysis of four normal beagles and four diabetic beagles. **i** Comprehensive Parkes error grid analysis of four normal beagles and four diabetic beagles.

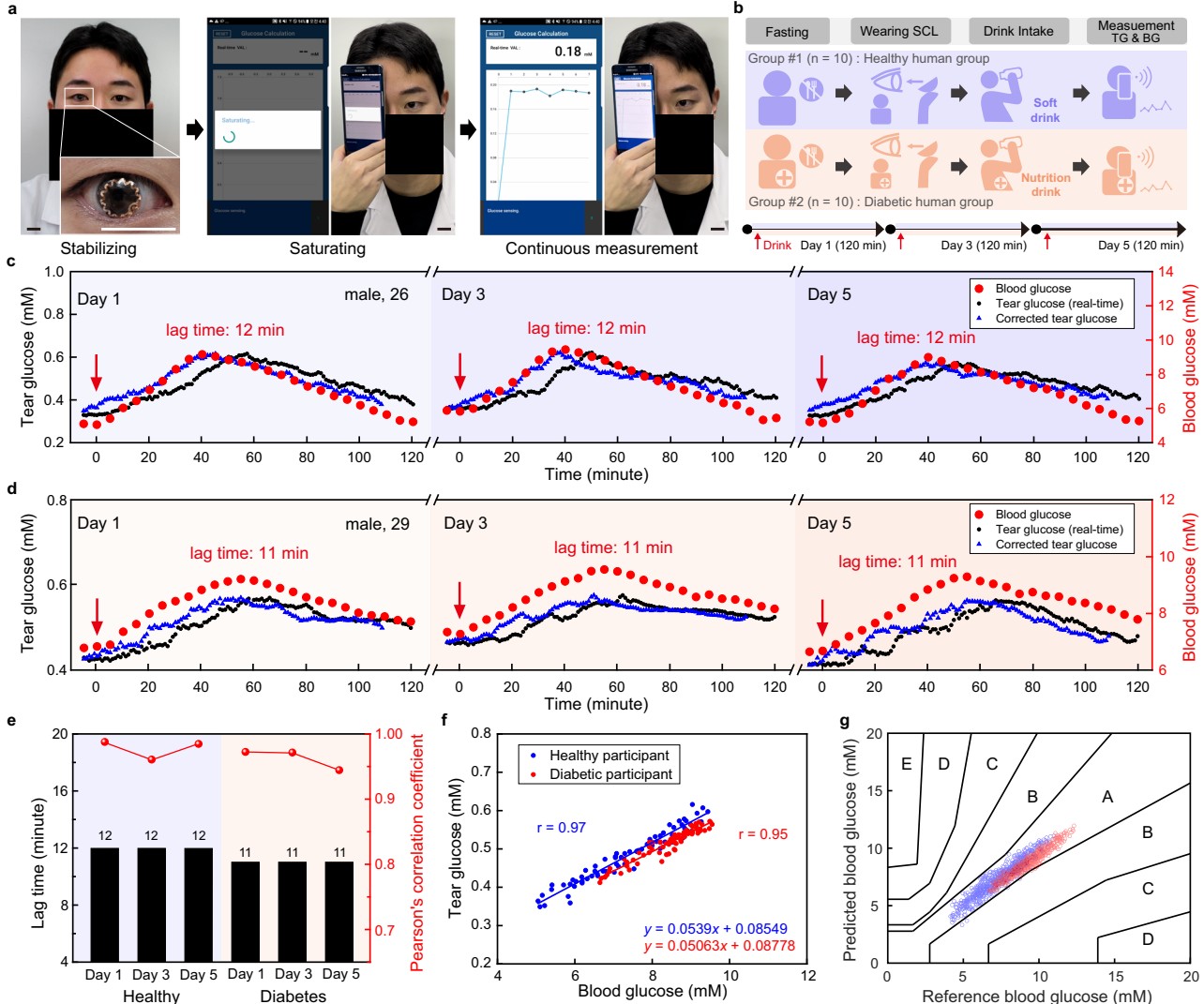

**Fig. 5 | Correlation analysis between tear glucose (TG) and blood glucose (BG) for healthy and diabetic human participants. a** Photographs of a human participant wearing smart contact lens (SCL) and TG measurement with SCL. Scale bars, 2 cm. Here, the participant's face was blocked using a black bar, except for the eyes, to infringe the portrait right. **b** Human study protocol for healthy and diabetic participants. **c** Representative real-time data of monitoring TG and BG level in a healthy participant after intake of a soft drink on day 1, 3, and 5. **d** Representative real-time data of monitoring TG and BG levels in a diabetic participant after intake of a nutrition beverage on day 1, 3, and 5. **e** Comparison of lag time in each human participant on day 1, 3, and 5. **f** Pearson's correlation analysis between TG and BG levels of each representative healthy and diabetic participant. **g** Comprehensive Parkes Error Grid analysis of ten healthy participants and ten diabetic participants.

We compared the ratio between TG and BG (TG/BG) obtained from Pearson's correlation analysis, resulting in values of 0.08771, 0.09394, 0.06037, and 0.07621 for normal rabbit, diabetic rabbit, normal beagle, and diabetic beagle, respectively. Rabbits exhibited slightly higher TG/BG ratios for both normal and diabetic groups compared to beagles, indicating differences in the partitioning ratio of glucose between tears and blood among species. The diabetic groups for both species also showed higher TG/BG ratios than the normal group, supporting the notion that diabetes can induce variations in the blood-tear barrier, affecting the transport of glucose across it[20,60,61].

**Human pilot study for correlation analysis between TG and BG**
To investigate the correlation analysis using SCL in humans, we conducted a human pilot study involving both ten healthy human participants and ten diabetic human participants (Supplementary Figs. 26 and 27). Figure 5a and Supplementary Movie 3 present a healthy 26-year-old male participant wearing our SCL and then measuring the TG level using a smartphone wirelessly. We have obtained consent for the individual to appear in the images. After wearing the SCL, under fasting conditions, healthy human participants were asked to drink a soft drink containing 54 g of sugar for OGTT. In the case of participants with diabetes, soft drinks were replaced with nutritional beverages, considering the potential adverse effects of soft drinks on diabetic patients during the experimental period based on physician's recommendation. After drinking, TG was measured continuously for 120 min, and the BG level also was measured using a glucometer with 5-min intervals. This testing procedure was conducted for 5 days on every other day, day 1, 3, and 5 (Fig. 5b). As shown in Fig. 5c, the representative result among the ten healthy participants, the fasting BG levels of a healthy participant were 5.05, 5.83, and 5.16 mM on day 1, 3, and 5, respectively, which are within the normal BG range. After drinking the soft drink, the BG levels increased up to 9.16, 9.44, and 8.99 mM on day 1, 3, and 5, respectively. And then gradually decreased, returning to baseline under the influence of general glucose homeostasis after 120 min. The fasting state TG levels of a normal participant were 0.329 mM, 0.376 mM, and 0.379 mM on day 1, 3, and 5, respectively, showing values similar to the reported TG range of human[34]. After consuming sugary drink, the TG levels increased to 0.616 mM,

0.621 mM, and 0.573 mM, and then decreased to 0.379 mM, 0.41 mM, and 0.405 mM. Although there were some individual differences, TG levels showed similar trends to BG levels in all participants (Supplementary Fig. 28). The fasting BG levels in a diabetic participant were slightly higher than those of normal participant, measuring 6.01, 6.63, and 6.18 mM on day 1, 3, and 5, respectively (Fig. 5d). After drinking the nutritional beverage, the BG levels increased up to 9.21, 9.55, and 9.29 mM on day 1, 3, and 5, respectively. The BG level did not return to the baseline within 120 min. This reflects a deficiency in the metabolic actions of insulin in diabetic patients[62]. Fasting TG levels in the diabetic participants were slightly higher than those of normal participants, showing 0.432, 0.474, and 0.423 mM on day 1, 3, and 5, respectively. The values increased to 0.57, 0.576, and 0.564 mM and then decreased to 0.499, 0.525, and 0.48 mM on day 1, 3, and 5, respectively. The TG levels followed the trend of BG levels with certain lag time, and the other nine diabetic participants exhibited similar results (Supplementary Fig. 29). Although the lag time varied randomly among individuals, the lag time, determined for one individual, was nearly remained constant on day 1, 3, and 5, regardless of whether the participants were healthy or diabetic (Fig. 5e and Supplementary Fig. 30), suggesting the personalized lag time may not be an instantly changing feature, rather be a maintaining feature over a certain period of time.

Figure 5f shows the results of Pearson's correlation analysis for a representative healthy and diabetic participant, which exhibited 0.97 and 0.95, respectively, as coefficient values. Such high levels indicate a positive correlation between TG and BG. The same analysis was performed for all remaining healthy and diabetic participants and the Pearson's correlation coefficient over 0.9 was observed in each individual (Supplementary Figs. 31, 32). The comprehensive Parkes Error Grid analysis revealed that the predicted BG data points, derived from the TG data using each individual's regression line equation, were all located within the A and B zones (Fig. 5g). This indicates that the prediction of BG level using our SCL was clinically accurate.

## Discussion

Commercialized continuous glucose monitoring devices (e.g., freestyle libre from Abbott) allow tight glycemic control for diabetes by measuring ISF glucose in a continuous way. The devices monitor ISF glucose via a subcutaneously injected needle, even though it is minimally invasive, not entirely non-invasive[63]. It can cause inflammation or bleeding which can deteriorate the accuracy of measured ISF glucose values via intrusion of BG[64], and foreign body response that may further induce the formation of fibrotic encapsulation around the loaded sensor, hindering the diffusion of glucose to the sensor surface and reducing the sensor responsivity[65]. In addition, the sensor body is worn on the skin surface using the adhesive that is known to often induce skin irritations, such as hypersensitivity reactions and contact dermatitis[66,67]. On the other hand, the SCL is fully non-invasive, eliminating the risk arising from invasiveness. It is unlikely to cause foreign body response, bleeding, or accompanying side effects. The SCL's intrinsic softness allows it to be worn and removed freely without causing any irritation to the eyes. Moreover, the SCL exhibited stretchable and biocompatible features designed to minimize eye stimulation while wearing. In this regard, SCL can be a promising wearable and non-invasive form factor to monitor glucose levels in tear fluid by incorporating glucose biosensors, providing the comfort for wearer. However, monitoring TG as a noninvasive alternative to BG has been raising questions regarding reliability, limiting their clinical use. Supplementary Table 1 summarizes those diverse opinions on the relationship between TG and BG. The most probable reason for the discrepancy is considered to be the differences in the tear collection methods. In particular, it has been reported that the glucose concentrations in mechanically stimulated tears are much higher than those in non-stimulated tears[20]. Another reason for the controversy may originate from the non-continuously obtained TG values. Since BG

fluctuates instantly and dynamically reflecting the status of carbohydrate metabolism and TG can follow these changes with a specific lag time, according to the diffusion mechanism of BG, it is important to measure both TG and BG continuously and frequently for an accurate correlation analysis. Since the lag time can vary among individuals and significantly impact the degree of correlation, it is important to determine the precise value of the lag time. However, previous studies presented a lack of measurement continuity, resulting in rough lag time estimations and inaccurate correlation analysis.

Thus, we developed wireless and soft SCL glucose biosensors and employed them to demonstrate the usefulness of TG as a noninvasive alternative to BG by addressing the above-mentioned concerns. First, we investigated the impact of mechanical stimulation on TG by wearing SCL and by involving external stimulation while wearing SCL, to mimic the possible disturbance that can happen in actual daily-life scenarios. It demonstrated that the developed SCL can disturb TG levels only at the moment of wearing it and even if there is external stimulation while wearing, it can soon recover to stable TG values within about 10 min, emphasizing the SCL can provide reliable and reproducible tear collection and TG values in continuous measurement settings, which can be useful for predicting BG. In addition, the SCL glucose biosensor possesses the capability of frequent and continuous measurement of TG, compared to previous TG biosensors (including SCLs), which offers more advanced monitoring performance of TG variations in real-time. It further allows an in-depth systematic evaluation of correlation analysis with BG and a more precise estimation of lag time which is a key factor for better prediction of BG using TG. We conducted in vivo experiments with both herbivores and omnivores by comparing normal and diabetic models, followed by human pilot studies with both healthy and diabetic participants. The study revealed that SCL can be used successfully for diagnosing diabetes and monitoring the status involving insulin intervention. Our SCL could measure low concentrations of TG levels continuously in both animal models (rabbit and beagle) and all human participants. Across all experimental results, we observed that the TG levels measured through SCL could reflect the variations in BG levels when the individual's lag time was determined accurately. After converting the measured TG data into predicted BG data using the regression line equation derived from lag time-based correlation analysis, Parkes Error Grid analysis was conducted. The resulting Parkes Error Grid analysis, all of the predicted BG data from animal models and human participants located in zone A and B, indicates SCL's high potential for clinical usability. Therefore, we identified that accurate estimation of the personalized lag time is crucial for the higher correlation coefficient between BG and TG, and it supports the reliability of our suggested methods on personalized lag time, considering various individuals' carbohydrate metabolism, towards better prediction of BG.

Furthermore, in this work, we made more efforts on in-depth validation of SCL glucose biosensors with human participants, by involving both diabetics and non-diabetics, towards practical utilization of SCL for continuous glucose monitoring in daily life. Figure 5f illustrates the correlation between TG and BG for a healthy human participant (male, 26). The regression line was derived from personalized lag time-based correlation analysis, and this equation can be used to convert TG data to BG data (x: BG, y: TG). If 0.5 mM of TG is measured, it can be converted to 7.69 mM of BG by substituting the TG value into this equation. Hence, the automated TG-BG level conversion is achievable for a single participant by simply entering the determined regression line equation into the application software of the mobile device (e.g., smartphone or tablet PC). To automatically convert TG to BG, the personalized lag time (for individuals) must first be investigated by monitoring BG and TG changes. From this initial investigation, we can set the personalized lag time and find a regression line using the measurement data points. Furthermore, we found the

individual's lag time (personalized lag time) remained steady rather than changed instantly via 5-day lasting human trials. In other words, the glucose diffusion rate from blood to tears may vary among individuals but can be a unique and lasting feature for each individual over a certain period, by reflecting the individual's health condition. Such variations of lag time among individuals were also observed between ISF glucose and BG[68–70], supporting our findings. Thus, further research could be followed in the future for a better understanding of the lag time between TG and BG over an extended period with a larger scale of human participants. In addition, future advancements may involve lag time-based correlation analysis through algorithm development. This algorithm could play a role in automatically establishing personalized lag time and regression line equations based on measured BG and TG data. The feasibility of this algorithm is likely to be realized after establishing a solid foundation through further extensive human studies. Subsequently, we can envision the development and utilization of an algorithm capable of automatically converting TG to BG[71].

Overall, our study attempted to obtain an in-depth understanding of the partitioning phenomenon of glucose between tears and blood using the developed SCL glucose biosensor by addressing, the existing questions on the reliability of TG monitoring for diabetes. Here we introduced the concept of "personalized lag time" and demonstrated its excellent performance for accurate prediction of BG, even with human diabetic participants, involving glucose intakes and insulin interventions, in consideration with practical utilization for patients. Such a result supports that our methodology on personalized lag time proposes the correct manner in understanding the behaviors of disease biomarkers in non-invasive biofluids and translating it into disease diagnosis and monitoring situations since all clinical criteria on disease are established on blood biomarker levels. The proposed method can be readily expanded to glucose and other biomarkers in other non-invasive biofluids, and other biomarkers in tear fluid, whose main mechanism of the presence is diffusion from blood. We believe it can further contribute to the significant advances for wearable biosensors towards commercialization, bringing better management of health status in daily life, based on user-friendly interfaces of non-invasive wearables.

## Methods

### Fabrication of the glucose biosensor

PDMS was spin-coated (4000 rpm, 40 s) on the $SiO_2$ wafer (Dasom RMS, Republic of Korea) as an adhesive layer which was followed by the attachment of a 25 μm-thick polyimide (PI) film. The Cr/Au (thickness, 10 nm/100 nm) was evaporated on PI film, and the photoresist (PR, S1818, Microchem Corp.) was spin-coated (3000 rpm, 40 s). For the patterning of the electrode, sequence of UV exposure (10 mW cm$^{-2}$, 18 s), development (AZ 300 K MIF, AZ electronic materials), and wet-etching using etchant (CE-905N, 10 s, TFA, 40 s, Transcene) were conducted. After removing the remained PR on the electrode with UV exposure (10 mW cm$^{-2}$, 5 min) and development, parylene (thickness, 500 nm) was coated as a passivation layer by chemical vapor deposition (CVD). To open the WE, RE/CE, and pads of the electrode, the PR was spin-coated (2000 rpm, 40 s) on the parylene layer and patterned by UV exposure and development process which was followed by dry-etching with oxygen plasma in reactive ion etching (RIE, 100 W, 40 s.c.c.m. of $O_2$, 150 s). The PB carbon ink (C2070424P2, Sun Chemical) and Ag/AgCl ink (Product #011464, ALS) were screen printed with the WE and RE/CE, through a screen-printing mask, respectively. Both inks were thermally cured at 60 °C for 20 min.

### Characterization of an electrochemical glucose biosensor

We immersed the glucose sensor in the PBS (pH 7.4, Sigma-Aldrich) filled in the beaker and glucose stock solution was added into the PBS solution to adjust the desired glucose concentration. For determining

the electrical characteristics of the glucose biosensor, cyclic voltammetry (CV) and chronoamperometry (CA) were performed using a potentiostat (PMC-CHS08A, Princeton Applied Research). CV curves were measured at each desired concentration of glucose in the potential window from 0.5 V to −0.5 V vs Ag/AgCl (scan rate, 50 mV s$^{-1}$). CA was measured under a constant potential of −0.1 V for 1 min at each concentrated glucose solution and the glucose biosensor was evaluated by detecting a change in electrical current. For the selectivity test, the sensor was tested in each solution with 0.04 mM ascorbic acid (Product #A92902, Sigma-Aldrich), 0.6 mM lactate (Product #PHR1113, Sigma-Aldrich), and 6 mM urea (Product #U5128, Sigma-Aldrich), respectively, because these are the well-known components of tear fluids. Each substance was dissolved into the artificial tears (NaCl, 5.5 g L$^{-1}$; KCl, 1.5 g L$^{-1}$) which were made by mixing diluted saline (Sigma-Aldrich) and KCl solution (Sigma-Aldrich). The measurement of the change in the current was conducted by CA using a potentiostat that applies the potential (vs Ag/AgCl) to the working electrode. For the pH stability test, the pH of PBS was adjusted by adding 0.1 M KCl, and it was checked continuously using a pH meter (Orion STAR A2116, Thermo Fisher). In the PBS with pH values that ranged from 6.4 to 7.6, the sensor detects various glucose concentrations (0.1-3 mM). The amperometric current was generated by the electrochemical reaction under the applied potential of −0.1 V (vs Ag/AgCl). For the stability test against long-term storage, the SCLs were stored in PBS for 0 to 21 days. The storage environment was about 25 °C in 10 mL of sterilized PBS (1 M, pH 7.4) which is similar to the condition of conventional contact lens storage.

### Fabrication of serpentine antenna and electromagnetic characterization

PDMS was spin-coated (4000 rpm, 40 s) on the $SiO_2$ wafer as an adhesive layer which was followed by the attachment of a 25 μm-thick PI film. The Cr/Cu layer (thickness, 10/3000 nm) was evaporated on the SiO2/PDMS/PI substrate. After PR spin-coating (2000 rpm, 30 s) on the substrate, the serpentine shape was patterned by photolithography procedure which was followed by the wet etching of the Cr/Cu layer by etchant (CE-905N, 10 s, Type 1, 30 s, Transene). The PI film that was not covered by the Cr/Cu was dry etched with RIE (200 W, 90 s.c.c.m of $O_2$, 55 s.c.c.m of Ar, 25 s.c.c.m of $SF_6$, 40 min). After wet-etching the metal mask on the remaining serpentine ring-shaped polyimide film, Cr/Au (thickness, 10 nm/100 nm) and Cr/Cu (thickness, 10 nm/5000 nm) were deposited through thermal evaporation and patterned in the form of the antenna contact pad and serpentine antenna, respectively. For the passivation of the antenna, 3 μm of the parylene layer was deposited and patterned by the photolithography process to open the area of the antenna contact pads. To open this area, dry etching was conducted through RIE (100 W, 40 s.c.c.m of $O_2$, 600 s) so that they could be electrically connected to the wireless circuit. A network analyzer (Rohde & Schwarz, ZNB 8) was used to characterize the properties of the wireless communication. The resonance frequency of the antenna was measured after integration in the lens with other components. The resonance frequency was measured wirelessly by the probe of the network analyzer, which was separated from the SCL by 5 mm.

### Integration of glucose sensor and wireless circuit into a soft contact lens

The fabricated antenna, glucose sensor, and capacitor (82 pF, Kyocera AVX) were electrically connected to an NFC chip (NHS 3152, NXP Semiconductors, thickness: 200 μm after polishing) with Au wire through a wire bonder (7KE, Westbond). Before the molding step, two electrodes of the glucose sensor were covered by a silicone elastomer ring and PI film as protection film. We put the integrated device into a mold of contact lenses and filled the mold with a precursor of silicone elastomer (Interojo®). The precursor of silicone elastomer in the mold

was cured at 70 °C for 5 h at a pressure of 313 kPa. After the molding process, the components of the SCL were completely embedded into the silicon elastomer. The PI protection film covering the sensor electrode was removed in order for the electrode to have direct contact with the glucose sensor with tear fluids. Then, 1.5 µL of the PBS solution with GOx (10 mg mL$^{-1}$) was dropped on a WE of the sensor. After the GOx solution dried at 4 °C for 4 h, we dropped a 2 µL mixture solution, composed of 1 wt% chitosan and 2 wt% acetic acid and dried it for 8 h. This procedure was applied twice to immobilize the GOx layer.

### Cell cytotoxicity assay with human cell lines

The human corneal epithelial cell line (HCE-2, #CRL-11135, clone 50.B1, ATCC) and human conjunctival epithelial cell line (HCECs, #CCL- 20-2, clone 1-5c-4, ATCC) were obtained from the American Type Culture Collection (ATCC, Manassas, VA, USA). The cells were cultured in DMEM/F-12 and RPMI 1640 medium (Gibco, MD, USA) supplemented with 10% heat-inactivated fetal bovine serum (FBS, Gibco), penicillin (50 U/ml) and streptomycin (50 µg/ml) (both from Welgene Inc. Gyeongsangbuk-do, Korea). Cells were plated with a density of $1 \times 10^5$ cells/ml in 96-well plates. The SCLs were sterilized with 70% ethanol for 3 min and washed three times in PBS for 2 min. Subsequently, the SCLs were incubated with culture medium containing 2% FBS at 37 °C for 24 h. The medium in which the SCL was immersed (SIM) was collected and stored at −80 °C until use. Each well was treated with 100 µl of SIM or media for 24 h. Cell viability was analyzed using protocol provided by Invitrogen™ (Invitrogen., MA, USA). Live cells were labeled with Calcein dye emitting a green fluorescence (ex/em-495/-515 nm), while dead cells were labeled with SYTOX DeepRed dye emitting a red fluorescence (ex/em maxima -660/628 nm). Followed by several washes, cells were observed and photographed with an Olympus fluorescence microscopy (Olympus BX51, Tokyo, Japan). Data were analyzed using GraphPad Prism version 8.0 software for Microsoft Windows 10. The results are presented as mean ± standard error of the mean. The unpaired student's T-test and the Mann-Whitney test were used to analyze the statistical differences between the two groups and the statistical significance was defined as $p > 0.05$.

### In vivo studies using rabbits and beagles

All in vivo tests using rabbits and beagles were conducted according to the guidelines of the Institute of Animal Care and Use Committee of Yonsei University (IACUC-202106-1276-04) and Daegu-Gyeongbuk Medical Innovation Foundation (DGMIF) (DGMIF-20071503-03, KMEDI-22012701-00). The Institute of Animal Care and Use Committee of Yonsei University and DGMIF are the ethics review committees. Male New Zealand white rabbits (3 kg, 4 months, Specific pathogen Free grade, four normal rabbits and four diabetic rabbits) and male beagle dogs (9 kg, 9 months, Conventional grade, four normal beagles and four diabetic beagles) were used. Beagle dogs were purchased from ORIENT BIO Inc (Korea, Republic of)., and New Zealand white rabbits were purchased from SAMTACO BIO KOREA (Korea, Republic of). We utilized animal models to investigate the operational performance of the SCLs according to the presence or absence of the disease, as well as its correlation with blood glucose levels. Therefore, gender differences in the disease were not considered. All experimental animals were housed at the K-MEDIHUB Preclinical Research Center located in Daegu, South Korea. The animal facility maintained controlled environmental conditions with relative humidity of 50 ± 5%, temperature of 23 °C ± 2.5 °C, and a 12-h light/dark cycle. Animals were housed in stainless steel cages equipped with automatic floor cleaning. During the breeding period, standard feed and clean water suitable for each species were provided freely. The sample size for each group was determined using the G*Power software (v. 3.1), and the animals used in the experiment were randomly divided into a control group or diabetes group (4 each) and experiments were performed.

### BG, TG, and TV measurement with conjunctival stimulation in normal rabbit

The rabbits fasted for 12 h and then they were anesthetized with a mixture of 2 mg kg$^{-1}$ of alfaxalone and 1 mg kg$^{-1}$ of xylazine. After applying SCL on the eye of the rabbits, wait for 10 min until the tear film stabilization. After stabilization, BG, TG, and TV were measured simultaneously every 5 min, 20 s, and 1 min, respectively. The conjunctival stimulation was conducted at specified times with 5, 20, and 35 min after starting measurements. The mechanical stimulation was conducted with von Frey filament (von Frey kit, Bioseb). The stimulation force of a von Frey filament was 1.4 gf. The TV measurement was conducted with strip meniscometry (SMTube, Echo Electricity).

### Simultaneous measurement of the glucose levels in both eyes of normal rabbits

For comparison of the TG concentrations in the right and left eyes, two SCLs were used. Each SCL was worn simultaneously on both eyes of a rabbit. The rabbit was anesthetized with a mixture of 2 mg kg$^{-1}$ of alfaxalone and 1 mg kg$^{-1}$ of xylazine before measuring the glucose in the tears. After wearing the SCLs, there were 10 min of TG stabilization time. After the stabilization of the TG levels, simultaneous TG measurements for both eyes were performed every 1 min for 20 min by SCLs.

### Oral glucose tolerance test (OGTT) in normal rabbits with SCL

OGTT was conducted with 12 h fasted New Zealand white rabbits (3.0 kg). The glucose solution used for oral administration was 50% (w/w) glucose-concentrated solution at 3.0 g kg$^{-1}$ of body weight. Then, the rabbits were anesthetized with a mixture of 2 mg kg$^{-1}$ of alfaxalone and 1 mg kg$^{-1}$ of xylazine. Using a glucose meter (ACCU-CHEK Performa, Roche), we measured BG by collecting blood via the auricular artery before the loading of the glucose solution and 10, 20, 30, 40, 50, 60, 70, 80, 90, 100, 110, and 120 min after the loading of the glucose solution. TG data were collected after wearing the SCL for 10 min for tear film stabilization. Throughout the experiment, TG was measured by SCL for 20 s every 1 min.

### Intravenous glucose tolerance test (IVGTT) in normal rabbits and beagles with SCL

An IVGTT was performed on four normal rabbits and four normal beagles. After anesthetizing (2 mg kg$^{-1}$ of alfaxalone and 1 mg kg$^{-1}$ of xylazine) of 12 h fasted rabbits and beagles, a glucose solution (50 wt%) with a dose of 1.0 g kg$^{-1}$ was injected into the rabbits and beagles via the marginal ear vein and foreleg vein, respectively. The BG of rabbits and beagles were measured via the auricular artery and the cephalic vein, respectively, before the injection of the glucose solution and 2, 5, 10, 15, 30, 45, 60, 75, 90, 105, and 120 min after the injection of the glucose solution. After 10 min of tear film stabilization time, TG was measured by SCLs for 20 s every 1 min throughout the experiment.

### Induction of diabetes in animal models and intravenous injection of glucose and insulin

For diabetic induction, a single shot of STZ injection was conducted. The rabbits and beagles had been fasting for 12 h and they were anesthetized lightly with a mixture of 2 mg kg$^{-1}$ of alfaxalone and 1 mg kg$^{-1}$ of xylazine. The 1% STZ solution was prepared by dissolving STZ (35 mg kg$^{-1}$ for beagles and 65 mg kg$^{-1}$ for rabbits) with 0.1 M citrate buffer and immediately injected into a marginal ear vein of rabbits or the foreleg vein of beagles. After STZ injection, BG measurement was conducted daily throughout the study. Rabbits or beagles with BG concentrations higher than 200 mg dl$^{-1}$ were considered as diabetic. With rabbits and beagles that had been fasting for 12 h and then anesthetized, a glucose solution (50 wt%) with a dose of 1.0 g kg$^{-1}$ was injected intravenously. Then, the rabbits and beagles were treated

with 0.5 U kg⁻¹ of insulin 45 min after the intravenous injection of glucose. BG was measured in the animals at times of 2, 5, 10, 15, 30, 45, 50, 55, 60, 75, 90, 105, and 120 min after the injection of the glucose solution. TG was measured for 20 s every 1 min by SCLs throughout the experiment.

## Histology

After the experiment was completed, the pancreas was harvested following euthanasia induction, and tissues were fixed using 10% neutral-buffered formalin. Paraffin sections were prepared by excising the medial part of the pancreas (body of the pancreas). Immunohisto-chemical staining for insulin and glucagon was performed to confirm changes in the beta cells of the islets of Langerhans. Following deparaffinization, sections were rehydrated gradually with ethanol and then antigen retrieval was performed for 40 min using citrate buffer (pH 6) at 95 °C. Then, sections were incubated with Monoclonal Anti-Insulin antibody produced in mouse IgG1 (Sigma-Aldrich, USA, Cat# I2018, diluted; 1:200) or rabbit anti-glucagon polyclonal antibody (MyBioSource, USA, Cat# MBS5314403, diluted; 1:200) for 90 min. After washing three times in PBS, mouse anti-rabbit IgG-HRP (Santa Cruz Biotechnology, USA, Cat# sc-2357, diluted; 1:400) and m-IgGκ BP-HRP (Santa Cruz Biotechnology, USA, Cat# sc-516102, diluted; 1:400) were incubated for 60 min. Hematoxylin was used for nuclear counterstain. Immunostained sections were analyzed under a microscope, and the area of the islets was measured using the Image J program (U.S. National Institutes of Health). The pancreas of a non-diabetic beagle was used as a control. The percentage of immune-positive cells for insulin and glucagon was calculated by dividing the number of immune-positive cells by the total number of cells in each of the 10 pancreatic islets.

## Human pilot study

The Institutional Review Board (IRB) of Yonsei University and Kyungpook National University Hospital approved the human pilot study protocol (7001988-202311-HR-1451-06, and KNUH 2022-11-003-005) which was conducted according to the Declaration of Helsinki, and all experimental procedures were performed with the informed consent of the participants. In addition, we obtained all participants' consent to use and publish information that identifies individuals, including indirect identifiers such as gender and age. Participants agreed to include images of their eyes with SCLs. At the end of our study, participants were compensated for their time. The human pilot study was conducted with ten healthy participants and ten diabetic participants. Before wearing SCL, this contact lens was rinsed using a commercial contact lens cleaning solution (Frenz-pro B5 solution, JK Pharmaceutical). After wearing SCL under fasting conditions, the participants waited for 10 min until the tear film was stabilized. Then, the baseline (i.e., the fasting states) of TG and BG were measured with SCL and a glucometer, respectively. Healthy participants were asked to drink soft drink containing 54 g of sugar, and diabetic participants were given nutrition beverages (NUCARE Glucose Plan, Daesang Wellife®, Seoul, Republic of Korea) containing 10 g of proteins, 7 g of dietary fiber, and 3 g of allulose for OGTT. After the intake, BG and TG were measured simultaneously for 120 min. 60 μL of blood was collected every 5 min for BG measurement, and TG was measured every minute for 20 s.

## Statistical analysis

All data were presented as mean ± standard error of the mean (SEM). Statistic calculations of $p$ value were performed using an open-source code of MATLAB.

## Reporting summary

Further information on research design is available in the Nature Portfolio Reporting Summary linked to this article.

## Data availability

The main data supporting the findings of this study are available within the paper and its Supplementary Information, and available at Figshare (https://doi.org/10.6084/m9.figshare.25323388). Any additional requests for information can be directed to, and will be fulfilled by the corresponding authors. Source data are provided with this paper.

## Code availability

The code used in this study is available as open-source on GitHub (https://github.com/4OH4/Parkes_EGA_MATLAB?tab=MIT-1-ov-file). The code was used with permission under copyright (c) 2016 4OH4.

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

## Acknowledgements

This work was supported by the Ministry of Science & ICT (MSIT), the Ministry of Trade, Industry and Energy (MOTIE), the Ministry of Health & Welfare, and the Ministry of Food and Drug Safety of Korea through the National Research Foundation (2023R1A2C2006257 (J.-U.P.) and 2021R1A2C2013939 (H.K.K)), ERC Program (2022R1A5A6000846) (J.-U.P.), and the Korea Medical Device Development Fund grant (RMS 2022-11-1209 / KMDF RS-2022-00141392) (J.-U.P.) and the grant of Korea Health Technology R&D Project through the Korea Health Industry Development Institute (KHIDI), funded by the Ministry of Health & Welfare (HR18C001202) (Jayoung K., Y.L.). Also, the authors thank the financial support of the Institute for Basic Science (IBS-R026-D1) (J.-U.P.) and Soo Bin Lim (Diabetes mellitus Center, Severance Hospital) for her help with human pilot study.

## Author contributions

Wonjung Park, Hunkyu Seo, and Jeongho Kim contributed the experiments, analyzed the data, and wrote the manuscript. Yeon-Mi Hong contributed to the human studies and conducted the data analysis. Hayoung Song and Byung Jun Joo fabricated the contact lenses. Sumin Kim participated in the beagle experiments and cell viability test. Enji Kim participated in the rabbit experiments. Che-Gyem Yae and Jonghwa Jin contributed to the healthy human studies. Jeonghyun Kim participated in the design of the wireless communication system. Joo-hee Kim contributed to the fabrication of wireless communication systems. Yong-ho Lee contributed to the planning of diabetic human studies and revised the manuscript. Jayoung Kim contributed to the design of the electrochemical biosensor and revised the manuscript. Hong Kyun Kim contributed to the planning of the project and the animal experiments. Jang-Ung Park oversaw all of the research phases and revised the manuscript. All the authors discussed and commented on the manuscript.

## Competing interests

The authors declare no competing interests.
