## [Peer Review File · Nature Communications]

REVIEWER COMMENTS

Reviewer #1 (Remarks to the Author):

Wonjung Park et described “Correlation analysis between tear glucose and blood glucose using a wireless smart contact lens” which is a very interesting, however some major issues should be addressed and it should be majorly revised before considering publication in Nature communications. Ethical declaration should be properly described although the authors provide IRB approval. Such as whether the study was conducted in adherence to the principles outlined in the Declaration of Helsinki. Also, the figure 5 include full image of the portrait, which seems to infringe the portrait right of him. The face should be covered by bars in order to block recognizing an individual per se for not violating portrait rights. Although the authors described cytotoxic test with human cell lines and histology to confirm diabetic model in Method, the results were not provided which would be very important to evaluate the safety of the CL and to confirm proper diabetic model. Last, there are very important issues that should be addressed in discussion. Concentration of tear glucose is very lower compared to concentration of blood glucose even if they correlate with blood glucose. Concentration of tear glucose looks like one-hundredth fold (1/100) of blood glucose concentration. How can it be clinically usable in clinics? Can the value of tear glucose conc. automatically convert to blood glucose conc. in your model for ordinary people to understand easily? The other important thing is that continuous blood glucose monitoring device with intradermal patch needle is already commercially available which can immediately and continuously transmit the glucose concentrations into the smart phone directly. What is the possible benefits of your model compared with the intradermal patch device for continuous blood glucose monitoring?

Reviewer #2 (Remarks to the Author):

The study aims to establish a correlation between tear glucose and blood glucose in terms of concentration and lag time in animal models and a pilot human trial.

Previous studies have widely reported device development and similar measurement results in animal models (rabbit). However, animal studies are not fully compatible with the human studies.

The data provided in the human study is not sufficient. The authors should provide data on both diabetic (n= >10) and healthy human subjects (n= >10) up to 14 days of continuous monitoring. The continuous measurement data should be compared to blood glucose measurements as well as a continuous glucose monitoring system (interstitial fluid).

[Response to Reviewer #1]

We appreciate the reviewer's thoughtful assessment and comments on our manuscript, and we welcome the opportunity to address and clarify the issues raised in the reviewer's report. Our responses to the comments on the report are as follows:

Comment 1: "Ethical declaration should be properly described although the authors provide IRB approval. Such as whether the study was conducted in adherence to the principles outlined in the Declaration of Helsinki."

Response to Comment 1:

Thanks for the reviewer's comment. Our study adhered to the standards outlined in the Declaration of Helsinki. A statement confirming adherence to the ethical declaration has been included in the Method section, along with additional information on the ethical aspects of human studies.

Revised manuscript (page 33-34, line 809-828)

The ~~protocol for the human pilot study was approved by the~~ Institutional Review Board (IRB) of Yonsei University and Kyungpook National University Hospital approved the human pilot study protocol (7001988-202311-HR-1451-06, and KNUH 2022-11-003-005) (~~7001988-202212-HR-1451-04~~) which was conducted according to the Declaration of Helsinki, and all experimental procedures were performed ~~done~~ with the informed consent of the participants. In addition, we obtained participants' consent to use and publish information that identifies individuals, including indirect identifiers such as gender and age. Participants agreed to include images of their eyes with SCLs. At the end of our study, participants were compensated for their time.

Comment 2: "Also, the figure 5 include full image of the portrait, which seems to infringe the portrait right of him. The face should be covered by bars in order to block recognizing an individual per se for not violating portrait rights."

Response to Comment 2:

We appreciate the valuable comment from the reviewer. As suggested, we have blocked the participant's face in Fig. 5a using a black bar, except for eyes. The same method has also been applied to block the face of the male participant in Supplementary Movie 3.

Revised manuscript (page 51-52)

Figure 5 | Correlation analysis between tear glucose (TG) and blood glucose (BG) for healthy and diabetic human participants. a, Photographs of a human ~~subject~~ participant wearing smart contact lens (SCL) and TG measurement with SCL. Scale bars, 2 cm. ~~Here, the participant's face was blocked using a black bar, except for eyes, to infringe the portrait right.~~

Revised Supplementary Movie

Supplementary Movie 3 | Human pilot trial of smart contact lens for simultaneous monitoring of tear glucose and blood glucose. Here, the participant's face was blocked using a black bar, except for eyes, to infringe the portrait right.

Comment 3: "Although the authors described cytotoxic test with human cell lines and histology to confirm diabetic model in Method, the results were not provided which would be very important to evaluate the safety of the CL and to confirm proper diabetic model."

Response to Comment 3:

We appreciate your feedback. The results of the cytotoxicity assay and histology, which are included in the supplementary information, were not sufficiently described in the manuscript. We apologize for any confusion this may have caused. *In-vitro* cytotoxicity tests were conducted for the smart contact lenses (SCLs) using human corneal (HCE-2, #CRL-11135, clone 50.B1, ATCC) and conjunctival (HCECs, #CCL-20-2, clone 1-5c-4, ATCC) cell lines. Cytotoxicity results using Cell Counting Kit-8 (CCK-8) showed cell viability of $97.87 \pm 1.77\%$ for HCE-2 and $95.28 \pm 1.67\%$ for HCECs, compared to controls, indicating no significant cytotoxicity. In addition, Calcein-AM/EthD-1 staining revealed no red-stained cells, confirming negligible toxicity. These results are presented in Supplementary Fig. 7 and 8. Additionally, to confirm the induction of diabetes in the animal models, histological analysis of the pancreatic islets was conducted after the completion of the *in-vivo* experiments using immunostaining for glucagon and insulin. The pancreatic islet area of diabetic rabbits and beagles were reduced by approximately 97.70% and 86.51%, respectively, compared to normal rabbits and beagles. These histological results indicate the successful induction of diabetes, and these results are presented in Supplementary Fig. 19 and 23. We have included these detailed explanations in the manuscript.

Revised manuscript (page 9-10, line 206-215)

The biocompatibility of ~~this~~ SCL ~~also~~ was ~~evaluated in~~ ~~confirmed by the cell viability test using~~ human corneal cells (HCE-2) and human conjunctival cells (HCECs), using a LIVE/DEAD assay kit and Cell Counting Kit-8 (CCK-8). Supplementary Fig. 7 shows that ~~resulting in~~ $97.87 \pm 1.77\%$ of HCE-2 and $95.28 \pm 1.67\%$ of HCECs did not exhibit significant cytotoxicity compared to controls. Calcein-AM/ethidium homodimer-1 double staining also revealed no significant dead cells, confirming negligible cytotoxicity (Supplementary Fig. 8), ~~respectively, as shown in the fluorescence microscope images in Supplementary Fig. 7 and 8.~~ These results ~~indicate suggest~~ that our SCL satisfied ~~did not exhibit significant cytotoxicity, and it satisfied~~ the cytotoxicity standard of over 80% ($\geq 80\%$) for medical devices (ISO 10993-5).

Revised manuscript (page 13-14, line 306-313)

To confirm the induction of diabetes in the rabbit models, we performed histological analysis of the pancreatic islets after completing the *in-vivo* experiments. Supplementary Fig. 19 displays immunostaining images for glucagon and insulin in the pancreatic islet. The pancreatic islet area of diabetic rabbits was reduced by approximately 97.70% compared to normal rabbits, indicating successful induction of diabetes in rabbit models. In detail, the normal rabbits had an area of $766.41 \pm 150.55 \mu\text{m}^2$, whereas the diabetic rabbits had an area of $17.63 \pm 8.70 \mu\text{m}^2$. ~~Also, images of the pancreatic islet with insulin and glucagon showed that the diabetic rabbits experienced severe losses of their pancreatic islet (Supplementary Fig. 19).~~

Revised manuscript (page 16, line 367-373)

Beagles ~~also~~ were ~~injected~~ ~~treated~~ with STZ ~~injections~~ to induce diabetes. Induction was ~~and~~ confirmed by ~~with~~ changes in body weights (Supplementary Fig. 22) and histological analysis after all experiments. Supplementary Fig. 23 shows a reduction in pancreatic islet area of 86.51% in diabetic beagles compared to normal beagles (normal beagles: $1506.51 \pm 271.80 \mu\text{m}^2$, diabetic beagles: $203.17 \pm 70.11 \mu\text{m}^2$), indicating successful induction of diabetes. ~~severe loss of pancreatic islet as diabetic rabbit model (Supplementary Fig. 22 and 23).~~

Revised manuscript (page 29-30, line 703-726)

~~For the cytotoxicity test of the SCL, †~~The human corneal epithelial cell line (HCE-2, #CRL-11135, clone 50.B1, ATCC) and human conjunctival epithelial cell line (HCECs, #CCL-20-2, clone 1-5c-4, ATCC) ~~were~~ obtained from the American Type Culture Collection (ATCC, Manassas, VA, USA). The cells were cultured in DMEM/F-12, ~~HEPES medium (Gibco)~~ and RPMI 1640 medium (Gibco, MD, USA); supplemented with 10% heat-inactivated fetal bovine serum (FBS, Gibco), penicillin (50 U/ml) and streptomycin (50 $\mu\text{g}/\text{ml}$) (both from Welgene Inc. Gyeongsangbuk-do, Korea). Cells were plated with a density of 1×10^5 cells/ml in 96-well plates. The SCLs were sterilized with 70% ethanol for 34 minutes; and washed ~~three~~ 3 times in PBS for 2 minutes.; Subsequently, the SCLs were incubated with ~~and eluted in a cell~~ culture medium containing 2% ~~FBS fetal bovine serum~~ at 37 °C for 24 hours. The medium in which the SCL was immersed (SIM) was collected and stored at -80°C until use. Each well was treated with 100 μl of SIM or media for 24 h. Cell viability was analyzed using protocol provided by Invitrogen™ (Invitrogen., MA, USA). Live cells were labeled with Calcein dye emitting a green fluorescence (ex/em~495/~515 nm), while dead cells were labeled with SYTOX DeepRed dye emitting a red fluorescence (ex/em maxima ~660/628 nm). Followed by several washes, cells were observed and photographed with an Olympus fluorescence microscopy (Olympus BX51, Tokyo, Japan). Data were analyzed using GraphPad Prism version 8.0 software for Microsoft Windows 10. ~~The cells that proliferated more than about 80% were incubated at 37 °C, in 5% CO₂ for 24 hours with the eluate. Then, the cytotoxicity assay (LIVE/DEAD Viability/Cytotoxicity kit, Invitrogen) and viability assay (Cell Counting Kit 8, Dojindo) were performed according to the manufacturer's protocol (ISO 10993-5). All data regarding cell viability data were measured by the GraphPad Prism version 8.0 software for Microsoft Windows 10 and~~ The results are presented ~~expressed~~ as mean \pm standard error of the mean.

Revised manuscript (page 33, line 806)

The percentage of immune-positive cells for insulin and glucagon was calculated by dividing the number of immune-positive cells by the total number of cells in each of the ~~12~~10 pancreatic islets.

Comment 4: "Last, there are very important issues that should be addressed in discussion. Concentration of tear glucose is very lower compared to concentration of blood glucose even if they correlate with blood glucose. Concentration of tear glucose looks like one-hundredth fold (1/100) of blood glucose concentration. How can it be clinically usable in clinics?"

Response to Comment 4:

We thank the reviewer for the comments. As mentioned by the reviewer, the concentration of tear glucose (TG) is significantly lower, compared to the blood glucose (BG) level. Clinical studies on the measurement of human TG concentration have shown that TG concentrations typically range from 0.18 to 0.7 mM (1), a concentration that cannot be measured using commercial blood glucose meters. The developed SCL in our study revealed a limit of detection (LOD) of 0.02 mM for glucose sensing, indicating its capability to detect low concentrations of TG. As indicated in the results section, our SCL could measure low concentrations of TG levels in both animal models (rabbits and beagles) and human participants. Furthermore, across all experimental results, it was observed that the TG levels measured through SCL followed the trend of BG levels (Pearson's correlation coefficient, $r > 0.9$). This suggests that even low concentrations of TG can continuously reflect changes in BG concentrations. For clinical usability, the most crucial aspect is the clinical accuracy of measurement results. The current criterion for determining the clinical accuracy of commercially available blood glucose meters is the Parkes error grid analysis. The Parkes error grid, also referred to as the consensus error grid, has been introduced as an accepted evaluation tool for BG monitoring systems in the ISO 15197:2013 guideline. To verify clinical accuracy, more than 95% of data points should be located in zones A and B of Parkes error grid. In our study, the TG data measured through SCL was converted into predicted BG using the regression line equation derived from correlation analysis. As a result, predicted BG data from animal models (rabbits and beagles) and human participants were all located in zones A and B, indicating the clinical accuracy. Therefore, we believe that the

predicted BG data derived from TG data holds a high potential for clinical usability. We included an additional description in the discussion section.

References:

1. Daum, K. M. & Hill, R. M. Human tear glucose. *Invest. Ophthalmol. Vis. Sci.* **22**, 509–514 (1982).

Revised manuscript (page 24, line 566-578)

~~Throughout the study, TG showed similar dynamic changes with BG, with a certain lag time obtained via the continuous measurement of TG and BG.~~ Our SCL could measure low concentrations of TG levels continuously in both animal models (rabbit and beagle) and all human participants. Across all experimental results, we observed that the TG levels measured through SCL could reflect the variations in BG levels when the individual's lag time was determined accurately. After converting the measured TG data into predicted BG data using the regression line equation derived from lag time-based correlation analysis, Parkes Error Grid analysis was conducted. The resulting Parkes Error Grid analysis, all of the predicted BG data from animal models and human participants located in zone A and B, indicates SCL's high potential for clinical usability. Therefore, we identified that accurate estimation of the personalized lag time is crucial for the higher correlation coefficient between BG and TG, and it supports the reliability of our suggested methods on personalized lag time, considering various individuals' carbohydrate metabolism, towards better prediction of BG.

Comment 5: "Can the value of tear glucose conc. automatically convert to blood glucose conc. in your model for ordinary people to understand easily?"

Response to Comment 5:

We would like to thank the reviewer for bringing this point to our attention. The objective of this study was to clarify the correlations between TG and BG in animals and humans using SCL. To achieve this, we measured BG and TG levels simultaneously in rabbits, beagles, and humans. Although there were variations in lag time and regression lines, all measured TG data showed a high correlation with BG levels. Based on the measured data, we can explain the process of converting TG into BG. For example, Figure R1 illustrates the correlation between TG and BG for a healthy human participant (male, 26). The regression line ($y = 0.0539x + 0.08549$) was derived from lag time-based correlation analysis, and this equation can be used to convert TG data to BG data (x : BG, y : TG). For instance, if 0.5 mM of TG is measured, it can be converted to 7.69 mM of BG by substituting the TG value into this equation (as indicated by the red dotted line). Hence, the automated TG-BG level conversion is achievable for a single individual by simply entering the determined regression line equation into the application of the portable device.

However, as observed in both animal and human studies, lag time and regression lines vary individually. To automatically convert TG to BG, the personalized lag time (for individuals) must first be investigated by monitoring BG and TG changes (as shown in Figure 5a). From this initial investigation, we can set the personalized lag time and find a regression line using the measurement data points. To facilitate these processes more efficiently, future advancements may involve lag time-based correlation analysis through algorithms. This algorithm could play a role in automatically establishing personalized lag time and regression line equations based on measured BG and TG data. The feasibility of this algorithm is likely to be realized after establishing a solid foundation through further extensive human studies. Subsequently, we can envision the development and utilization of an algorithm capable of automatically converting TG to BG. We included an additional description in the discussion section.

Figure R1 | Conversion of the TG level to the BG level in healthy human participant.

Revised manuscript (page 24-25, line 579-591)

Furthermore, in this work, we made more efforts on in-depth validation of SCL glucose biosensors with human participants, by involving both diabetics and non-diabetics, towards practical utilization of SCL for continuous glucose monitoring in daily life. Fig. 5f illustrates the correlation between TG and BG for a healthy human participant (male, 26). The regression line was derived from personalized lag time-based correlation analysis, and this equation can be used to convert TG data to BG data (x : BG, y : TG). If 0.5 mM of TG is measured, it can be converted to 7.69 mM of BG by substituting the TG value into this equation. Hence, the automated TG-BG level conversion is achievable for a single participant by simply entering the determined regression line equation into the application software of the mobile device (e.g., smartphone or tablet PC). To automatically convert TG to BG, the personalized lag time (for individuals) must first be investigated by monitoring BG and TG changes. From this initial investigation, we can set the personalized lag time and find a regression line using the measurement data points.

Revised manuscript (page 25, line 598-604)

In addition, future advancements may involve lag time-based correlation analysis through algorithm development. This algorithm could play a role in automatically establishing personalized lag time and regression line equations based on measured BG and TG data. The feasibility of this algorithm is likely to be realized after establishing a solid foundation through further extensive human studies. Subsequently, we can envision the development and utilization of an algorithm capable of automatically converting TG to BG.

Comment 6: "The other important thing is that continuous blood glucose monitoring device with intradermal patch needle is already commercially available which can immediately and continuously transmit the glucose concentrations into the smart phone directly. What is the possible benefits of your model compared with the intradermal patch device for continuous blood glucose monitoring?"

Response to Comment 6:

Thanks for the insightful comment. As the author mentioned, the intradermal patch needle device is already commercially available for continuous glucose monitoring. This device utilizes interstitial fluid (ISF) for continuous glucose monitoring. The measurement of glucose through the intradermal patch needle device offers advantages in terms of continuity and minimal invasiveness, making it practical alternative to conventional glucometer measurement methods. However, there are still limitations that need to be addressed. Firstly, this patch device is not entirely a non-invasive method. Since the ISF is located beneath the skin, microneedles of a few millimeters in length should be inserted into the skin (1). Due to this invasive characteristic, inflammation or bleeding may occur, which causes distortion of glucose concentration in the ISF (2). Furthermore, a foreign body response occurs after loading the device. The foreign body response induces the formation of fibrotic encapsulation around the loaded sensor, hindering the diffusion of glucose to the sensor surface and reducing the sensor responsivity (3). Secondly, skin irritation frequently occurs at the attachment site. For reliable glucose sensing with intradermal patch devices, a robust attachment to the skin is essential. The adhesive layer used for device attachment is known to often induce skin irritations, such as hypersensitivity reactions and contact dermatitis (4). Lastly, wearing an intradermal patch causes discomfort in daily life. Intradermal patch device is impossible to detach it according to the user's situation, as it must remain continuously attached throughout the period of use. In a survey conducted on intradermal patch devices, the predominant reason reported for discontinuing the use of intradermal patch devices was the discomfort (5).

In contrast, SCL is fully non-invasive, eliminating the risk arising from invasiveness. Our SCL has virtually no chance of causing foreign body response, bleeding, or side effects. Moreover, SCL's intrinsic softness allows it to be worn and removed freely without causing any irritation to the eyes. In this way, SCLs have an advantage over commercialized intradermal patch devices in terms of non-invasiveness, and convenience. We expect that these advantages will make the SCL platform more convenient to use in everyday life compared to commercial intradermal patch devices. We have added a comparison with the intradermal patch to the discussion section.

References:

1. Wang, Y., Wu, Y. & Lei, Y. Microneedle-based glucose monitoring: a review from sampling methods to wearable biosensors. *Biomater. Sci.* **11**, 5727–5757 (2023).
2. Friedel, M. et al. Opportunities and challenges in the diagnostic utility of dermal interstitial fluid. *Nat. Biomed. Eng.* **7**, 1541–1555 (2023).
3. McClatchey, P. M. et al. Fibrotic Encapsulation Is the Dominant Source of Continuous Glucose Monitor Delays. *Diabetes* **68**, 1892–1901 (2019).

4. Messer, L. H., Berget, C., Beatson, C., Polsky, S. & Forlenza, G. P. Preserving Skin Integrity with Chronic Device Use in Diabetes. *Diabetes Technol. Ther.* **20**, S2-54 (2018).
5. Wong, J. C. et al. Real-Time Continuous Glucose Monitoring Among Participants in the T1D Exchange Clinic Registry. *Diabetes Care* **37**, 2702–2709 (2014).

Revised manuscript (page 21-23, line 493-561)

Commercialized continuous glucose monitoring devices (e.g., freestyle libre from Abbott) allow tight glycemic control for diabetes by measuring ISF glucose in a continuous way. The devices monitor ISF glucose via a subcutaneously injected needle, even though it is minimally invasive, not entirely non-invasive⁶³. It can cause inflammation or bleeding which can deteriorate the accuracy of measured ISF glucose values via intrusion of BG⁶⁴, and foreign body response that may further induce the formation of fibrotic encapsulation around the loaded sensor, hindering the diffusion of glucose to the sensor surface and reducing the sensor responsivity⁶⁵. In addition, the sensor body is worn on the skin surface using the adhesive that is known to often induce skin irritations, such as hypersensitivity reactions and contact dermatitis^{66,67}. On the other hand, the SCL is fully non-invasive, eliminating the risk arising from invasiveness. It is unlikely to cause foreign body response, bleeding, or accompanying side effects. The SCL's intrinsic softness allows it to be worn and removed freely without causing any irritation to the eyes. Moreover, the SCL exhibited stretchable and biocompatible features designed to minimize eye stimulation while wearing. In this regard, SCL can be a promising wearable and non-invasive form factor to monitor glucose levels in tear fluid by incorporating glucose biosensors, providing the comfort for wearer. However, monitoring TG as a noninvasive alternative to BG has been raising questions regarding reliability, limiting their clinical use. Supplementary Table 1 summarizes those diverse opinions on the relationship between TG and BG. The most probable reason for the discrepancy is considered to be the differences in the tear collection methods. In particular, it has been reported that the glucose concentrations in mechanically stimulated tears are much higher than those in non-stimulated tears²⁰. Another reason for the controversy may originate from the non-continuously obtained TG values. Since BG fluctuates instantly and dynamically reflecting the status of carbohydrate metabolism and TG can follow these changes with a specific lag time, according to the diffusion mechanism of BG, it is important to measure both TG and BG continuously and frequently for an accurate correlation analysis. Since the lag time can vary among individuals and significantly impact the degree of correlation, it is important to determine the precise value of the lag time. However, previous studies presented a lack of measurement continuity, resulting in rough lag time estimations and inaccurate correlation analysis.

~~For the clinical use of wearable devices utilizing non-invasive body fluids, a biological understanding of the correlation between blood and other body fluids must be preceded. However, in the case of diagnostic devices using tears, the correlation with blood is not sufficiently established, limiting their clinical use. For example, Supplementary Table 1 shows that their opinions are still not unified even though there have been many studies that used various methods in attempts to elucidate the correlation between TG and BG. The most probable reason for the discrepancy between these studies is thought to be the differences in the methods that were used to collect the tears. In particular, it has been reported that the glucose concentrations in mechanically stimulated tears are much higher than those in non-stimulated tears²⁰. Another reason for the controversy is the non-continuous measurements of TG. Since BG fluctuates according to body conditions and TG follows these changes with a specific lag time, it is important to measure both TG and BG continuously for an accurate correlation analysis. Since the lag time varies among individuals and can significantly impact the degree of correlation, it is especially crucial to accurately determine the precise value of the lag time. However, previous studies lacked measurement continuity, resulting in rough lag time assumptions and inaccurate correlation analysis.~~

Thus, we developed wireless and soft SCL glucose biosensors and employed them to demonstrate the usefulness of TG as a noninvasive alternative to BG by addressing the above-mentioned concerns. First, we investigated the impact of mechanical stimulation on TG by wearing SCL and by involving external stimulation while wearing SCL, to mimic the possible disturbance that can happen in actual daily-life scenarios. It demonstrated that the developed SCL can disturb TG levels only at the moment of wearing it and even if there is external stimulation while wearing, it can soon recover to stable TG values within about 10 minutes, emphasizing the SCL can provide reliable and reproducible tear collection and TG values in continuous measurement settings, which can be useful for predicting BG. In addition, the SCL glucose biosensor possesses the capability of frequent and continuous measurement of TG, compared to previous TG biosensors (including SCLs), which offers more advanced monitoring performance of TG variations in real-time. It further allows an in-depth systematic evaluation of correlation analysis with BG and a more precise estimation of lag time which is a key factor for better prediction of BG using TG. ~~In this study, we introduced the SCL as a promising method for the correlation analysis between tears and blood. The developed SCL biosensor devices exhibited soft, stretchable, and biocompatible features designed to minimize the~~

~~stimulation of the eye. We demonstrated that SCL disturbs TG levels only at the time of wearing, indicating that SCL can provide non-stimulating tear collection with TG monitoring and continuous measurement of TG. In addition, we verified the effect of external stimulation on the TG levels with normal rabbits wearing the SCL, and we found that this stimulation increased TG levels and impaired the correlation with blood, suggesting that mechanical stimulation may be the main reason for the controversy of correlation between TG and BG. Also, TG values were recovered soon to stable values within about 10 minutes, again emphasizing the need for continuous measurement of TG for better prediction of BG.~~

Revised manuscript (page 41-42, line 994-1003)

63. Wang, Y., Wu, Y. & Lei, Y. Microneedle-based glucose monitoring: a review from sampling methods to wearable biosensors. *Biomater. Sci.* **11**, 5727–5757 (2023).
64. Friedel, M. et al. Opportunities and challenges in the diagnostic utility of dermal interstitial fluid. *Nat. Biomed. Eng.* **7**, 1541–1555 (2023).
65. McClatchey, P. M. et al. Fibrotic Encapsulation Is the Dominant Source of Continuous Glucose Monitor Delays. *Diabetes* **68**, 1892–1901 (2019).
66. Messer, L. H., Berget, C., Beatson, C., Polsky, S. & Forlenza, G. P. Preserving Skin Integrity with Chronic Device Use in Diabetes. *Diabetes Technol. Ther.* **20**, S2-54 (2018).
67. Wong, J. C. et al. Real-Time Continuous Glucose Monitoring Among Participants in the T1D Exchange Clinic Registry. *Diabetes Care* **37**, 2702–2709 (2014).

[Response to Reviewer #2]

We appreciate the reviewer's thoughtful assessment and comments on our manuscript, and we welcome the opportunity to address and clarify the issues raised in the reviewer's report. Our responses to the comments on the report are as follows:

Comment 1: "Previous studies have widely reported device development and similar measurement results in animal models (rabbit). However, animal studies are not fully compatible with the human studies. The data provided in the human study is not sufficient. The authors should provide data on both diabetic (n = >10) and healthy human subjects (n = >10) up to 14 days of continuous monitoring. The continuous measurement data should be compared to blood glucose measurements as well as a continuous glucose monitoring system (interstitial fluid)."

Response to Comment 1:

Thanks for the insightful comment. As mentioned by the reviewer, animal studies are not fully compatible with human studies, and more data from human studies need to be provided. The editor had additional comments regarding the reviewer's suggestions, specifically recommending a reduction in the suggested 14 days of measurement period. Instead, the editor suggested conducting blood glucose (BG) and tear glucose (TG) measurements on alternating days for up to 5 days, at a minimum, testing on day 1, 3, and 5. In response to the opinions of both the reviewer and editor, we conducted additional human studies with healthy human participants (n = 10) and diabetic human participants (n = 10) on day 1, 3, and 5.

The results of the correlation analysis between TG and BG in both groups indicated a consistently high Pearson's correlation coefficient, exceeding 0.9 for all individuals. Additionally, the comprehensive Parkes Error Grid analysis for both healthy and diabetic participants revealed that all data points were located within the A and B zones. These results collectively demonstrate a clear correlation between the measured TG levels and BG levels in both healthy and diabetic participants. Detailed explanations have been added to the overall manuscript.

Revised manuscript (page 3, line 60-61)

This demonstration considers individual differences and has been applied successfully to both non-diabetic and diabetic humans, as well as in animal models ~~and human participants in this study~~, resulting in high correlation.

Revised manuscript (page 6-7, line 133-146)

Lastly, such correlation analysis using the developed SCL was further demonstrated with healthy and diabetic human participants, and different animal species (e.g., herbivorous rabbits and omnivorous beagles) with and without diabetes, ~~and to the same species without diabetes~~ for comparative analysis. ~~There were differences in the lag times among individual animals, different species, and the presence or absence of disease that could affect the degree of correlation.~~ For the better prediction of BG, ~~the above-mentioned concept of personalized lag time has been utilized successfully for better BG prediction for~~ in humans, rabbits, and beagles which resulted in a clear demonstration of the correlation between TG and BG. ~~models, and this approach was further demonstrated with healthy human participants.~~ Conducting a comprehensive analysis across various species adds further significance for the demonstration of the correlation between TG and BG. To the best of our knowledge, this is the first in-depth study to use SCL for examining the correlation between TG and BG across various species including humans, rabbits, and beagles with diabetes, in comparison with healthy groups as control ones.

Revised manuscript (page 18-21, line 428-490)

To investigate the correlation analysis using SCL in humans, we conducted a human pilot study involving both ten healthy human participants and ten diabetic human participants (Supplementary Fig. 26 and 27). Fig. 5a and Supplementary Movie 3 present a healthy 26-year-old male participant wearing our SCL and then measuring the TG level using a smartphone wirelessly. After wearing the SCL, under fasting conditions, healthy human participants were asked to drink a soft drink containing 54 g of sugar for OGTT. In the case of participants with diabetes, soft drinks were replaced with nutritional beverages, considering the potential adverse effects of soft drinks on diabetic patients during the experimental period based on physician's recommendation. After drinking, TG was measured continuously for 120 minutes, and the BG level also was measured using a glucometer with 5-minute intervals. This testing procedure was conducted for 5 days on every other day, day 1, 3, and 5 (Fig. 5b). As shown in Fig. 5c, the representative result among the ten healthy participants, the fasting BG levels of a healthy participant were 5.05, 5.83, and 5.16 mM on day 1, 3, and 5, respectively, which are within the normal BG range. After drinking the soft drink, the BG

levels increased up to 9.16, 9.44, and 8.99 mM on day 1, 3, and 5, respectively. And then gradually decreased, returning to baseline under the influence of general glucose homeostasis after 120 minutes. The fasting state TG levels of a normal participant were 0.329 mM, 0.376 mM, and 0.379 mM on day 1, 3, and 5, respectively, showing values similar to the reported TG range of human³⁴. After consuming sugary drink, the TG levels increased to 0.616 mM, 0.621 mM, and 0.573 mM, and then decreased to 0.379 mM, 0.41 mM, and 0.405 mM. Although there were some individual differences, TG levels showed similar trends to BG levels in all participants (Supplementary Fig. 28). The fasting BG levels in a diabetic participant were slightly higher than those of normal participant, measuring 6.01, 6.63, and 6.18 mM on day 1, 3, and 5, respectively (Fig. 5d). After drinking the nutritional beverage, the BG levels increased up to 9.21, 9.55, and 9.29 mM on day 1, 3, and 5, respectively. The BG level did not return to the baseline within 120 minutes. This reflects a deficiency in the metabolic actions of insulin in diabetic patients⁶². Fasting TG levels in the diabetic participant were slightly higher than those of normal participant, showing 0.432, 0.474, and 0.423 mM on day 1, 3, and 5, respectively. The values increased to 0.57, 0.576, and 0.564 mM and then decreased to 0.499, 0.525, and 0.48 mM on day 1, 3, and 5, respectively. The TG levels followed the trend of BG levels with certain lag time, and the other nine diabetic participants exhibited similar results (Supplementary Fig. 29). The OGTT was conducted with the real-time measurement of TG and BG in five participants (Supplementary Fig. 26). After wearing the SCL, the participants were asked to drink a sugary beverage (soft drink with 54 g of sugar). Subsequently, TG was measured continuously for 120 minutes, and the BG level also was measured using a glucometer with 5-minute intervals. As shown in Fig. 5b, the representative result among the five participants, the fasting state showed a BG level of 4.27 mM, which is in the normal BG range. After the consumption of the beverage, the BG levels spiked up to 8.82 mM and then decreased gradually to 5.61 mM at the end of the 120-minute measurement via glucose homeostasis. The TG levels were 0.178 mM, 0.46 mM, and 0.287 mM during the fasting state, at the maximum value, and at the endpoint of OGTT, respectively. Although there were differences in the baseline TG and BG levels between participants due to individual differences in glucose absorption from the beverages and metabolic actions of insulin, similar trends in TG and BG were observed in the other four participants (Supplementary Fig. 27). Similar to previous investigations in the animal models, the lag times were also present (15, 12, 12, 8 and 12 minutes in the five participants), with the highest Pearson's correlation coefficients (Fig. 5e). Although the lag time varied randomly among individuals, the lag time, determined for one individual, was nearly remained constant on day 1, 3, and 5, regardless of whether the participants were healthy or diabetic (Fig. 5e and Supplementary Fig. 30), suggesting the personalized lag time may not be an instantly changing feature, rather be a maintaining feature over a certain period of time.

Fig. 5f shows the results of Pearson's correlation analysis for a representative healthy and diabetic participant, which exhibited 0.97 and 0.95, respectively, as coefficient values. Such high levels indicate a positive correlation between TG and BG. The same analysis was performed for all remaining healthy and diabetic participants and the Pearson's correlation coefficient over 0.9 was observed in each individual (Supplementary Fig.31 and 32). The comprehensive Parkes Error Grid analysis revealed that the predicted BG data points, derived from the TG data using each individual's regression line equation, were all located within the A and B zones (Fig. 5g). This indicates that the prediction of BG level using our SCL was clinically accurate. Fig. 5d presents a plot of the comprehensive Pearson's correlation data of these five human participants with a coefficient value of 0.954, which is high enough to demonstrate the positive correlation between TG and BG. All data points were within zone A of the Parkes Error Grid, indicating that the prediction of BG level using our SCL was clinically appropriate (Fig. 5e).

Revised manuscript (page 23-24, line 561-566)

~~For an in-depth investigation of the correlation between TG and BG, w~~We conducted *in-vivo* experiments with both herbivores and omnivores by comparing normal and diabetic models, followed by human pilot studies with both healthy and diabetic participants. The study revealed that SCL can be used successfully for diagnosing diabetes and monitoring the status involving insulin intervention.

Revised manuscript (page 25, line 591-598)

Furthermore, we found the individual's lag time (personalized lag time) remained steady rather than changed instantly via 5-day lasting human trials. In other words, the glucose diffusion rate from blood to tears may vary among individuals but can be a unique and lasting feature for each individual over a certain period, by reflecting the individual's health condition. Such variations of lag time among individuals were also observed between ISF glucose and BG^{68,69,70}, supporting our findings. Thus, further research could be followed in the future for a better understanding of the lag time between TG and BG over an extended period with a larger scale of human participants.

Revised manuscript (page 25-26, line 605-623)

Overall, our study attempted to obtain an in-depth understanding of the partitioning phenomenon of glucose between tears and blood using the developed SCL glucose biosensor by addressing the existing questions on the reliability of TG monitoring for diabetes. Here we introduced the concept of ‘personalized lag time’ and demonstrated its excellent performance for accurate prediction of BG, even with human diabetic participants, involving glucose intakes and insulin interventions, in consideration with practical utilization for patients. Such a result supports that our methodology on personalized lag time proposes the correct manner in understanding the behaviors of disease biomarkers in non-invasive biofluids and translating it into disease diagnosis and monitoring situations since all clinical criteria on disease are established on blood biomarker levels. The proposed method can be readily expanded to glucose and other biomarkers in other non-invasive biofluids, and other biomarkers in tear fluid, whose main mechanism of the presence is diffusion from blood. We believe it can further contribute to the significant advances for wearable biosensors towards commercialization, bringing better management of health status in daily life, based on user-friendly interfaces of non-invasive wearables. ~~controversies of correlation between TG and BG, and insights towards accurate BG estimation using TG. Based on our results, a tear-based diagnostic device can reliably measure the glucose levels in the body continuously with comfort in daily settings. Thus, it is reasonable to expect that considerable progress will be made in the development of non-invasive biofluid-based wearable devices.~~

Revised manuscript (page 34, line 817-828)

The human pilot study was conducted with ~~five~~ ten healthy participants and ten diabetic participants. Before wearing SCL, this contact lens was rinsed using a commercial contact lens cleaning solution (Frenz-pro B5 solution, JK Pharmaceutical). After wearing SCL under fasting conditions, the participants waited for 10 minutes until the tear film was stabilized. Then, the baseline (i.e., the fasting states) of TG and BG were measured with SCL and a glucometer, respectively. Healthy participants were asked to drink soft drink containing 54 g of sugar, and diabetic participants were given nutrition beverages (NUCARE Glucose Plan, Daesang Wellife®, Seoul, Republic of Korea) containing 10 g of proteins, 7 g of dietary fiber, and 3 g of allulose for OGTT. After the intake ~~of a sugary beverage (500 ml of a soft drink with 54 g of sugar)~~, BG and TG were measured simultaneously for 120 minutes. 60 µL of blood was collected ~~BG was measured~~ every 5 minutes for BG measurement, and TG was measured every minute for 20 seconds.

Revised manuscript (page 41-42, line 992-993, 104-1009)

62. ElSayed, N. A. et al. 2. Classification and Diagnosis of Diabetes: Standards of Care in Diabetes—2023. *Diabetes Care* **46**, S19–S40 (2022).
68. Basu, A. et al. Time Lag of Glucose From Intravascular to Interstitial Compartment in Type 1 Diabetes. *J Diabetes Sci Technol* **9**, 63–68 (2014).
69. Chlup, R. et al. Glucose concentrations in blood and tissue - a pilot study on variable time lag. *Biomed Pap Med Fac Univ Palacky Olomouc Czech Repub* **159**, 527–534 (2015).
70. Basu, A. et al. Time Lag of Glucose From Intravascular to Interstitial Compartment in Humans. *Diabetes* **62**, 4083–4087 (2013).

Figure 5 | Correlation analysis between tear glucose (TG) and blood glucose (BG) for healthy and diabetic human participants. **a**, Photographs of a human subject participant wearing smart contact lens (SCL) and TG measurement with SCL. Scale bars, 2 cm. Here, the participant's face was blocked using a black bar, except for eyes, to infringe the portrait right. **b**, Human study protocol for healthy and diabetic participants. **bc**, Representative real-time data of monitoring TG and BG level in a human healthy subject participant after intake of a soft drink sugary beverage on day 1, 3, and 5. **ed**, Representative real-time data of monitoring TG and BG level in a diabetic participant after intake of a nutrition beverage on day 1, 3, and 5. **e**, Comparison of lag time and Pearson's correlation coefficient in each human subject. **de**, Comprehensive Pearson's correlation analysis between TG and BG level in five human subjects. **f**, Comparison of lag time in each human participant on day 1, 3, and 5. **e**, Comprehensive Parkes error grid analysis in five human subjects. **f**, Pearson's correlation analysis between TG and BG levels of each representative healthy and diabetic participant. **g**, Comprehensive Parkes Error Grid analysis of ten healthy participants and ten diabetic participants.

Revised Supplementary Information (page 29-30)

Human subject	Eye	Wearing image
1	Left		Right	2	Left		Right	3	Left		Right	4	Left		Right	5	Left		Right	
Human	Wearing image	Human	Wearing image
1		6	2		7	3		8	4		9	5		10	
Supplementary Figure 26. ~~Five Images of ten healthy participants human-subjects wearing the SCL. smart contact lens.~~

Revised Supplementary Information (page 31-32)

Human	Wearing image	Human	Wearing image
1		6	
2		7	
3		8	
4		9	
5		10	

Supplementary Figure 27. Images of ten diabetic participants wearing the SCL. Real-time measurement of tear glucose and blood glucose of each human after intake of sugary beverage.

Revised Supplementary Information (page 33)

Supplementary Figure 28. Real-time data of monitoring TG and BG level in healthy participants (ii-x) after intake of a soft drink on day 1, 3, and 5.

Revised Supplementary Information (page 34)

Supplementary Figure 29. Real-time data of monitoring TG and BG level in diabetic participants (ii-x) after intake of a nutrition beverage on day 1, 3, and 5.

Revised Supplementary Information (page 35)

Supplementary Figure 30. The lag time and Pearson's correlation coefficient on day 1, 3, and 5. a, Healthy participants (n = 10). b, Diabetic participants (n = 10)

Revised Supplementary Information (page 36)

Supplementary Figure 31. Pearson's correlation analysis between TG and BG in healthy participants (ii-x), measured on day 1, 3, and 5.

Revised Supplementary Information (page 37)

Supplementary Figure 32. Pearson's correlation analysis between TG and BG in diabetic participants (ii-x), measured on day 1, 3, and 5.

[Additional Modifications]

Modification 1:

The title of the paper has been slightly modified.

Revised Manuscript (Page 1):

In-depth Correlation analysis between tear glucose and blood glucose using a wireless smart contact lens

Revised Supplementary Information (Page 1):

In-depth Correlation analysis between tear glucose and blood glucose using a wireless smart contact lens

Modification 2:

All authors agree to the addition of three authors by considering their extensive work during the revision process.

Revised Manuscript (Page 1):

Wonjung Park^{1,3†}, Hunkyu Seo^{1,3†}, Jeongho Kim^{54†}, **Yeon-Mi Hong^{1,3}**, Hayoung Song^{1,3}, Byung Jun Joo^{1,3}, Sumin Kim^{1,3}, Enji Kim^{1,3}, **Che-Gyem Yae⁶**, Jeonghyun Kim⁷⁵, **Jonghwa Jin⁸**, Joohee Kim^{96*}, **Yong-ho Lee^{10,11,12*}**, Jayoung Kim^{137*}, Hong Kyun Kim^{5,6,144,9,10*}, Jang-Ung Park^{1,2,3,48*}.

Revised Manuscript (Page 43-44, line 1029-1039):

~~Wonjung Park~~, ~~Hunkyu Seo~~, and ~~Jeongho Kim~~ contributed the experiments, analyzed the data, and wrote the manuscript. **Yeon-Mi Hong contributed to the human studies and conducted the data analysis.** ~~Hayoung Song~~ and ~~Byung Jun Joo~~ fabricated the contact lenses. ~~Sumin Kim~~ participated in the beagle experiments and cell viability test. ~~Enji Kim~~ participated in the rabbit experiments. **Che-Gyem Yae and Jonghwa Jin contributed to the healthy human studies.** ~~Jeonghyun Kim~~ participated in the design of the wireless communication system. ~~Joohee Kim~~ contributed to the fabrication of wireless communication systems. **Yong-ho Lee contributed to the planning of diabetic human studies and revised the manuscript.** ~~Jayoung Kim~~ contributed to the design of the electrochemical biosensor and revised the manuscript. ~~Hong Kyun Kim~~ contributed to the planning of the project and the animal experiments. ~~Jang-Ung Park~~ oversaw all of the research phases and revised the manuscript. All the authors discussed and commented on the manuscript.

Revised Supplementary Information (Page 1):

Wonjung Park^{1,3†}, Hunkyu Seo^{1,3†}, Jeongho Kim^{54†}, **Yeon-Mi Hong^{1,3}**, Hayoung Song^{1,3}, Byung Jun Joo^{1,3}, Sumin Kim^{1,3}, Enji Kim^{1,3}, **Che-Gyem Yae⁶**, Jeonghyun Kim⁷⁵, **Jonghwa Jin⁸**, Joohee Kim^{96*}, **Yong-ho Lee^{10,11,12*}**, Jayoung Kim^{137*}, Hong Kyun Kim^{5,6,144,9,10*}, Jang-Ung Park^{1,2,3,48*}.

Modification 3:

Acknowledgment has been revised.

Revised Manuscript (Page 43, line 1019-1026):

This work was supported by the Ministry of Science & ICT (MSIT), the Ministry of Trade, Industry and Energy (MOTIE), the Ministry of Health & Welfare, and the Ministry of Food and Drug Safety of Korea through the National Research Foundation (2023R1A2C2006257 and 2021R1A2C2013939), Nano Material Technology Development Program (2021M3D1A2049914), ERC Program (2022R1A5A6000846, 2020R1A5A1019131), the Technology Innovation Program (20013621, Center for Super Critical Material Industrial Technology), and the Korea Medical Device Development Fund grant (RMS 2022-11-1209 / KMDF RS-2022-00141392) **and the grant of Korea Health Technology R&D Project through the Korea Health Industry Development Institute (KHIDI), funded by the Ministry of Health & Welfare (HR18C001202).** Also, the authors thank the financial support by the ~~Samsung Research Funding & Incubation Center of Samsung Electronics (SRFC-TC2003-03)~~ and the Institute for Basic Science (IBS-R026-D1) and Soo Bin Lim (Diabetes mellitus Center, Severance Hospital) for her help with human pilot study.

Modification 4:

A reporting summary statement has been included.

Revised Manuscript (Page 35, line 839-841):

Reporting summary

Further information on research design is available in the Nature Research Reporting Summary linked to this article.

Modification 5:

A code availability statement has been included.

Revised Manuscript (Page 35, line 843-845):

Code availability

The custom code used in this study is available from the corresponding author upon reasonable request.

REVIEWERS' COMMENTS

Reviewer #1 (Remarks to the Author):

The manuscript was revised as appropriately and is ready to publish.

Reviewer #2 (Remarks to the Author):

The authors have addressed all my comments.